# Persuasive Prediction via Decision Calibration

## Abstract

Bayesian persuasion, a central model in information design, studies how a sender, who privately observes a state drawn from a prior distribution, strategically sends a signal to influence a receiver's action. A key assumption is that both sender and receiver share the precise knowledge of the prior. Although this prior can be estimated from past data, such assumptions break down in high-dimensional or infinite state spaces, where learning an accurate prior may require a prohibitive amount of data. In this paper, we study a learning-based variant of persuasion, which we term *persuasive prediction*. This setting mirrors Bayesian persuasion with large state spaces, but crucially does not assume a common prior: the sender observes covariates $X$, learns to predict a payoff-relevant outcome $Y$ from past data, and releases a prediction to influence a population of receivers. To model rational receiver behavior without a common prior, we adopt a learnable proxy: *decision calibration*, which requires the prediction to be unbiased conditioned on the receiver's best response to the prediction. This condition guarantees that myopically responding to the prediction yields no swap regret. Assuming the receivers best respond to decision-calibrated predictors, we design a provably efficient algorithm that learns a decision-calibrated predictor within a randomized predictor class that optimizes the sender's utility. In the commonly studied single-receiver case, our method matches the utility of a Bayesian sender who has full knowledge of the underlying prior distribution. Finally, we extend our algorithmic result to a setting where receivers respond stochastically to predictions and the sender may randomize over an infinite predictor class.

## 1 Introduction

The strategic disclosure of information to influence downstream decisions—commonly studied under the *information design* literature (see e.g., Bergemann et al. (2023))—has become a central topic in economic theory. A foundational model in this area is *Bayesian persuasion*, introduced by Kamenica & Gentzkow (2011), which formalizes how a *sender*, who privately observes an underlying state unknown to the *receiver*, can send a signal to shape the receiver's belief and ultimately influence their chosen action. Crucially, the sender is endowed with the *power of commitment*—the ability to credibly commit to a signaling scheme in advance. Based on the committed signaling scheme, the receiver updates their beliefs upon receiving the signal and selects an action accordingly. Both receivers have utility functions that depend on the state and the receiver's action, but these utilities are typically misaligned, so that the sender needs to strategically design their scheme.

A key assumption underpinning Bayesian persuasion—and much of information design—is the existence of a *common prior*: a probability distribution over the state space that both the sender and receiver have precise knowledge about. The most obvious source for such a probability distribution is past data. However, in many real-world environments, the state space may be high-dimensional or even infinite, which makes it challenging to estimate an accurate prior distribution from finite data. A natural example is the following setting of prediction for decision making.

**Persuasive Prediction Problem.** The data $(x, y) \in \mathcal{X} \times \mathcal{Y}$ are drawn from a joint distribution $\mathcal{D}$. The sender observes a realization $x$ of $\mathcal{X}$, and seeks to predict the unobserved outcome $y$. Both the sender's and the receiver's utilities depend on the receiver's action and the unknown outcome $Y$, but

not directly on $X$. Upon observing $x$, the sender provides a prediction $f(x)$, which acts as a signal to inform the receiver's decision. In the language of Bayesian persuasion, the sender's observed state corresponds to the posterior distribution $\mathbb{P}[Y \mid X = x]$, and the prior is the marginal distribution over such posterior distributions, which is effectively $\mathcal{D}$. However, when $\mathcal{X}$ is high-dimensional or infinite, learning the prior $\mathcal{D}$ from finite past data can be infeasible.

This persuasive prediction framework captures a wide range of real-world scenarios where the sender observes high-dimensional covariates describing the state of the world and aims to shape individual decisions toward a socially desirable outcome. For example, a small community bank may partner with a fintech platform that outputs predicted default probabilities based on high-dimensional financial and behavioral data. However, the incentives of the two parties are not fully aligned: the fintech platform benefits from increasing loan origination volume and thus may have incentives to produce predictions that make approvals more likely, whereas the bank remains cautious and primarily uses these predictions to safeguard against excessive default risk.

When data are insufficient to recover the full prior, persuasive prediction presents two central challenges: *How should receiver behavior be modeled in the absence of a common prior? And can the sender, using only finite data, achieve utility comparable to that of a fully informed Bayesian sender with exact knowledge of the distribution $\mathcal{D}$?*

## 1.1 OUR RESULTS AND TECHNIQUES

In persuasive prediction, a sender learns a predictor $f$ from past data such that upon observing $X = x$, they will send the prediction $f(x)$ to a population of receivers, who will then select their actions. The sender's goal is to optimize their utility that depends jointly on the unknown outcome $Y$ and the joint action chosen by the receivers. Our results establish a connection between (Bayesian) persuasion and *decision calibration* (Noarov et al., 2023; Zhao et al., 2021), which allows a learning-based approach for modeling incentives without requiring full knowledge of the prior.

**Behavioral Modeling via Decision Calibration.** Informally, a (possibly randomized) predictor $f: \mathcal{X} \to \mathcal{Y}$ is *decision-calibrated* if, for every receiver with utility function $v_i$ and every action $a_i$,

$$\mathbb{E}_{f,(X,Y) \sim \mathcal{D}} \left[ Y - f(X) \mid \arg\max_a v_i(f(X), a) = a_i \right] = 0. \tag{1}$$

Intuitively, decision calibration captures a natural notion of credibility: conditioned on any event defined by the receiver's best response, the predictor $f(X)$ must be an unbiased estimate of the true outcome $Y$. A predictor is approximately decision-calibrated if condition equation 1 holds up to a small additive error. We show that myopically best responding to an approximately decision-calibrated, receivers obtain low swap regret (Lemma 2.1). This motivates a clean behavioral assumption that receivers best respond to approximately decision calibrated predictions.

**Efficient Optimal Persuasive Prediction.** Suppose the sender is allowed to use a stochastic predictor $f \in \Delta(\mathcal{H})$, that is, a distribution over deterministic predictors in some finite class $\mathcal{H}$. Our first main result is a statistically efficient algorithm that learns a predictor $f$ that optimizes the sender's utility within the class of decision-calibrated predictors in $\Delta(\mathcal{H})$. The core technical idea is to formulate the learning problem as a zero-sum game between a *min player*, who updates the predictor, and an *max player*, who identifies the most violated calibration constraint. Simulating no-regret dynamics between the two players yields a minimax equilibrium, which recovers the optimal decision-calibrated predictor. The number of required samples scales polynomially in the number of receiver actions and the dimension of $\mathcal{Y}$, and is independent of the size $|\mathcal{X}|$. Our algorithm is also *oracle-efficient* in the sense that it runs in polynomial time when given access to an ERM oracle over $\mathcal{H}$.

**Matching the Bayesian Benchmark.** In the special case of a single receiver, we show that the sender utility achieved by our algorithm matches that of a fully informed Bayesian sender who is restricted to sending signals induced by the same class of decision-calibrated predictors, even though our sender only has access to a finite dataset that is far from sufficient to approximate $\mathcal{D}$.

**Extension to Quantal Responses.** Finally, we extend our results to settings where receivers perform *quantal responses*-—that is, their action choices follow a softmax distribution rather than deter-

ministic best responses. This extension models scenarios where receiver behavior is stochastic and not perfectly rational (McKelvey & Palfrey, 1995). Under this setting, we also provide an efficient algorithm for learning the approximately optimal decision-calibrated predictor for sender utility, and can also handle infinite hypothesis classes $\mathcal{H}$, provided it has bounded covering numbers.

## 1.2 RELATED WORK

The work most closely related to ours is Feng & Tang (2025), which studied the problem of selecting an optimal calibrated predictor to maximize the learner's utility. Part of their result was based on Jain & Perchet (2024), who established the connection between online calibration and Bayesian persuasion. Notably, Feng & Tang (2025) assumed a finite context space $\mathcal{X}$ and that the learner knows the distribution $\mathcal{D}$ over $\mathcal{X} \times \mathcal{Y}$. We also consider a selection problem: choosing an optimal decision-calibrated predictor to maximize the learner's utility, but in a more challenging setting: (i) we focus on a prior-free model, where the sender does not know $\mathcal{D}$ and the context space $\mathcal{X}$ is rich enough so that learning the conditional distribution $D_{\mathcal{Y}|x}$ for arbitrary $x$ is infeasible, and (ii) we allow the outcome space to extend beyond the binary case.

Our work is also conceptually related to recent work on prior-free mechanisms. Lin & Li (2024) studied Bayesian persuasion without knowing the prior. They showed that, under certain regularity conditions, it is possible to learn an approximately optimal signaling scheme by first estimating the prior from the data and then solving the persuasion problem with the estimated prior. Their approach is infeasible in our setting since we do not assume $\mathcal{D}$ to be learnable with a finite sample. Camara et al. (2020); Collina et al. (2024) studied a repeated Principal-Agent problem between a pair of long-lived Principal and Agent in an adversarial setting where there is no prior distribution. To address the challenges of the online setting, they impose additional rationality assumptions on the agent's behavior. In contrast, we make no such assumptions and only require that the agent (receiver in our case) follows the (smoothed) best response. Due to space limitations, we include additional related work in the Appendix D.

## 2 MODEL AND PRELIMINARIES

**Predictors** We consider the prediction task over the data domain $\mathcal{X} \times \mathcal{Y}$, where the data is drawn from a distribution $\mathcal{D}$. Here, $\mathcal{X}$ is a rich feature space, and $\mathcal{Y} = [-1, 1]^d$ is the outcome space. We define $\mathcal{H} = \{h \mid h : \mathcal{X} \to \mathcal{Y}\}$ as a hypothesis class of *deterministic* predictors. For any $h \in \mathcal{H}$ and $x \in \mathcal{X}$, $h(x)$ is interpreted as a prediction of the conditional mean $\mathbb{E}[Y \mid X = x]$. We use $h(x)_j$ and $y_j$ to denote the $j$-coordinate of the predicted and true outcome vectors, respectively.

In this paper, we consider a more general setting where the goal is to learn a *randomized* predictor $f \in \Delta(\mathcal{H})$, representing a distribution over $h \in \mathcal{H}$. This aligns with the standard information design literature, where a sender typically transmits a randomized signal to the receiver to achieve higher utility. We assume no direct access to the full distribution $\mathcal{D}$; instead, we seek to learn $f$ from $n$ i.i.d. samples drawn from $\mathcal{D}$, which we denote as dataset $D$.

**Receivers' Behavior Model** We consider $N$ receivers who make decisions based on the prediction $h(x)$. For each $i \in [N]$, receiver $i$ has a finite action set $\mathcal{A}_i$. Without loss of generality, we assume $|\mathcal{A}_i| = m$ for all $i \in [N]$, since any smaller set can be augmented with dummy actions to reach size $m$. Receiver $i$'s utility function is denoted as $v_i(a, y)$, where $v_i : \mathcal{A}_i \times \mathcal{Y} \longrightarrow [0, 1]$.

We assume that $v_i$ is linear and Lipschitz continuous in the outcome $y$.

**Assumption 2.1** (Linearity and $L$-Lipschitzness). *For any $i \in [N]$, and $a \in \mathcal{A}_i$, the utility function $v_i(a, y)$ is linear in $y$, and satisfies $|v_i(a, y_1) - v_i(a, y_2)| \leq L\|y_1 - y_2\|_\infty$.*

Next, we define the receiver's decision rule given the prediction $h(x)$. A natural rule is to treat the prediction as accurate and respond optimally to it.

**Definition 2.1** (Strict Best Response). *For any $i \in [N]$, receiver $i$, given utility function $v_i$, strictly best responds to the prediction $h(x)$ by choosing:*

$$b_i(h(x), a_i) = \begin{cases} 1 & \text{if } a_i = \arg\max_{a_i' \in \mathcal{A}_i} v_i(a_i', h(x)), \\ 0 & \text{otherwise}. \end{cases}$$

*Here, $b_i(h(x), a_i)$ represents the probability that receiver $i$ takes action $a_i$ given the prediction $h(x)$.*

**Decision Calibration**    We aim to design $f$ such that receivers experience no regret when best responding to it. This mirrors the setting in standard Bayesian persuasion, where, given a known prior, the sender recommends an action through a signal, and the receiver's best response—after Bayesian updating—is to follow that recommendation. As we do not assume a known prior, we leverage the notion of *decision calibration* (Noarov et al., 2023; Zhao et al., 2021), which has been shown to provide similar no-regret guarantees for receivers. Specifically, it ensures that receivers have no incentive to deviate from the recommended action, whether by swapping actions or by acting as if their utility were that of another receiver. Moreover, it can be shown that any approximately decision-calibrated predictor can be post-processed into an approximately fully calibrated predictor that induces the same receiver's behavior. We will show that fully calibrated predictors correspond to signaling schemes in Bayesian persuasion with a known prior (Lemma 4.1). A similar observation was made in (Jain & Perchet, 2024) for online calibrated predictions. But they made finite-state Bayesian persuasion assumption, which we do not require. For these reasons, we adopt decision calibration as a desirable property that the predictor $f$ should satisfy. We provide a detailed discussion of the no-regret guarantees in Appendix F, and their connection to Bayesian persuasion in Section 4.

We now formally define decision calibration as follows.

**Definition 2.2** (Decision Calibration). *A randomized predictor $f \in \Delta(\mathcal{H})$ is said to be perfectly decision calibrated if*

$$\mathrm{DecCE}(f) := \max_{i \in [N]} \max_{j \in [d]} \max_{a \in \mathcal{A}_i} \left| \mathbb{E}_{h \sim f} \mathbb{E}_{(x,y) \sim \mathcal{D}} [(y_j - h(x)_j) \cdot b_i(h(x), a)] \right| = 0.$$

*Moreover, $f$ is said to be $\epsilon$-decision calibrated if $\mathrm{DecCE}(f) \leq \epsilon$.*

A decision-calibrated predictor ensures that receivers have no incentive to deviate from best responding to the prediction; that is, receivers cannot achieve higher utility by swapping their chosen action with another action. Similar guarantees have been established by Noarov et al. (2023); Roth & Shi (2024), and we provide a variant tailored to our setting of randomized predictor in the distributional setting. We formally define swap regret as follows:

**Definition 2.3** (Swap Regret). *We say that a predictor $f$ achieves $\epsilon$-swap regret if, for any receiver $i \in [N]$, mapping function $\phi : A \to A$,*

$$\mathbb{E}_{h \sim f} \mathbb{E}_{\mathcal{D}} \left[ \sum_a v_i(\phi(a), y) \cdot b_i(h(x), a) \right] \leq \mathbb{E}_{h \sim f} \mathbb{E}_{\mathcal{D}} \left[ \sum_a v_i(a, y) \cdot b_i(h(x), a) \right] + \epsilon.$$

**Theorem 2.1** (No Swap Regret via Decision Calibration). *If a predictor $f$ is $\epsilon$-decision calibrated, then it has at most $2L|A|\epsilon$-swap regret.*

We further show that decision calibration can guarantee other forms of regret, such as *type regret*, which ensures that receivers have no incentive to pretend to be another receiver, as well as combinations of swap and type regret. These results demonstrate that decision calibration is a strongly compatible with receivers' incentive. We provide detailed statements and proofs in Appendix F.

We consider randomized predictors in $\Delta(\mathcal{H})$ that has decision calibration error bounded by some target level $\gamma$ and assume that such predictors are not vacuous.

**Assumption 2.2** (Feasibility). *There is a randomized predictor $f \in \Delta(\mathcal{H})$ that $\mathrm{DecCE}(f) \leq \gamma$.*

Note that Assumption 2.2 is mild since as long as $\mathcal{H}$ contains all deterministic constant predictors (or a discretized cover thereof), it necessarily includes a decision-calibrated predictor, specifically the constant predictor $h(x) = \mathbb{E}[Y]$, which is fully calibrated.

**Sender's Objective**    The sender does *not* have direct access to the data distribution $\mathcal{D}$, but has examples drawn from $\mathcal{D}$. The sender's utility depends on the outcome $y$ and the joint action of all receivers, given by $u : \mathcal{A} \times \mathcal{Y} \to \mathbb{R}$ where $\mathcal{A} := \mathcal{A}_1 \times \cdots \times \mathcal{A}_N$. Without loss of generality, we assume the sender's utility is bounded by 1, i.e. $u(a, y) \in [0, 1]$ for any $a \in \mathcal{A}$ and $y \in \mathcal{Y}$. Given any prediction $h(x)$, for any $\boldsymbol{a} = (a_1, \cdots, a_N) \in \mathcal{A}$, the probability that receivers play joint action $\boldsymbol{a}$ is $b(h(x), \boldsymbol{a}) := \prod_{i=1}^{N} b_i(h(x), a_i)$.

The sender's goal is to maximize their expected utility subject to a $\gamma$-decision calibration constraint. Formally, the sender's optimization problem is

$$\max_{f} \mathbb{E}_{h\sim f}\mathbb{E}_{(x,y)\sim\mathcal{D}}\left[\sum_{\boldsymbol{a}\in\mathcal{A}} u(\boldsymbol{a},y)b(h(x),\boldsymbol{a})\right] \quad \text{s.t.} \quad \text{DecCE}(f)\le\gamma. \tag{2}$$

We denote the optimal objective value of Eq. (2) by $\text{OPT}(\mathcal{H},\mathcal{D},\gamma)$.

## 3 A MINIMAX APPROACH FOR EFFICIENT PERSUASIVE PREDICTION

In this section, we present an efficient algorithm `PerDecCal` (Algorithm 1) for persuasive prediction, which learns an approximately optimal solution to the constrained optimization problem Eq. (2) from data when $|\mathcal{H}|$ is finite. We argue that finite hypothesis classes are already powerful: in finite-state Bayesian persuasion, a simple finite class suffices, as randomization over it can achieve the same sender utility as the full space of continuous signaling schemes. A detailed discussion is provided in Appendix E

We begin by stating the theoretical guarantee achieved by `PerDecCal`.

**Theorem 3.1.** *Suppose* `PerDecCal` *runs for* $T = O(\log(Nmd)/\epsilon^4)$ *rounds and is given a dataset* $D$ *drawn i.i.d. from* $\mathcal{D}$ *of size* $n \ge O(\frac{\log(|\mathcal{H}|Ndm/\delta)}{\epsilon^4})$. *With probability at least* $1-\delta$, *the output predictor* $\hat{f}$ *that satisfies*

1. $\text{DecCE}(\hat{f}) \le \gamma + \epsilon$.

2. *Suppose the receivers play strict best response to* $\hat{f}$. *Then the receivers obtain swap regret bounded by* $2mL(\gamma+\epsilon)$. *The sender achieves* $\epsilon$-*optimal utility:*
$$\mathbb{E}_{h\sim f}\mathbb{E}_{\mathcal{D}}[u(\boldsymbol{a},y)\cdot b(h(x),\boldsymbol{a})] \ge \text{OPT}(\mathcal{H},\mathcal{D},\gamma) - \epsilon.$$

Theorem 3.1 shows that, with enough sample size, our proposed algorithm `PerDecCal` learns a predictor $\hat{f}$ that achieves nearly optimal utility compared to the best in-class $\gamma$-decision-calibrated predictor, while ensuring that its decision calibration error exceeds $\gamma$ by at most $\epsilon$, and this ensures that the receivers have no regret best responding to the predictions. It establishes a bi-criteria optimization: instead of requiring the predictor to be exactly $\gamma$-decision calibrated, we allow slight violations, which increase the swap regret by at most $2mL\varepsilon$. Furthermore, we prove a lower bound showing that it is statistically infeasible to learn a near-optimal predictor within the class of exactly $\gamma$-decision calibrated predictors. We defer the details to Appendix G.

`PerDecCal` follows a minimax-based approach. Specifically, we introduce Lagrangian variables and reformulate the original problem as a minimax game. We then apply an oracle-efficient algorithm to compute an approximate equilibrium of this game, which yields a near-optimal solution to the original problem Eq. (2). We now present the details and analysis of `PerDecCal`, with full proofs provided in Appendix H.

**Lagrangian and Minimax Game** As a standard technique in optimization theory, the constrained optimization problem can be equivalently written in its Lagrangian form, which can be interpreted as a minimax game. Specifically, we introduce the Lagrangian as follows:

$$\min_{f\in\Delta\mathcal{H}} \max_{\lambda\in\mathbb{R}_+^{2Nmd}} \mathcal{L}_{\mathcal{D}}(f,\lambda) := -\mathbb{E}_{h\sim f}\mathbb{E}_{\mathcal{D}}\left[\sum_{\boldsymbol{a}\in\mathcal{A}} u(\boldsymbol{a},y)\cdot b(h(x),\boldsymbol{a})\right]$$
$$+ \sum_{s\in\{+,-\}}\sum_{i=1}^{N}\sum_{j=1}^{d}\sum_{a_i\in A_i} \lambda_{s,i,j,a_i} s(\mathbb{E}_f\mathbb{E}_{\mathcal{D}}[(h(x)_j - y_j)\cdot b_i(h(x),a_i)] - \gamma). \tag{3}$$

By the folklore result in optimization theory (Boyd & Vandenberghe (2004)), the minimax solution of Eq. (3) coincides with the optimal solution of Eq. (2). After introducing the Lagrange multipliers, Eq. (3) can be viewed as a minimax game, where the minimization player is the predictor $f$, and the maximization player is the Lagrangian multiplier $\lambda$.

**Best Response vs. No Regret Dynamics** Viewing the problem as a minimax game, we consider solving it using *Best Response vs. No Regret* (BRNR) dynamics, where the min player $f$ plays a best response to the current $\lambda$, and the max player updates $\lambda$ according to a no-regret algorithm. Freund & Schapire (1996) showed that when both players achieve low regret, the average of their plays converges to an approximate equilibrium. Since both $\Delta(\mathcal{H})$ and $\mathbb{R}_+^{2Nmd}$ are convex spaces, we can apply their result once we establish sublinear regret for both players.

For the min player $f$, this is straightforward because $\mathcal{L}(f, \lambda)$ is linear in the randomized predictor $f$. As a result, the best response to any fixed $\lambda$ is achieved by a deterministic predictor, that is, $\arg\min_{f\in\Delta(\mathcal{H})} \mathcal{L}(f, \lambda) = \arg\min_{h\in\mathcal{H}} \mathcal{L}(h, \lambda)$. However, since the max player's strategy space $\mathbb{R}_+^{2Nmd}$ is unbounded, designing a no-regret algorithm for the max player is non-trivial, as standard regret-minimization algorithms typically require bounded decision spaces.

**Bounded Minimax Games** To design a no-regret algorithm for the max player $\lambda$, we first restrict the Lagrangian variables to be bounded. Specifically, we consider $\lambda \in \Lambda = \{\lambda' | \lambda' \in \mathbb{R}_+^{2Nmd}, \|\lambda\|_1 \leq C\}$. When the $\ell_1$ norm of $\lambda$ is bounded, it becomes straightforward to design a no-regret algorithm for the domain $\Lambda$. We define the $C$-bounded minimax games as follows:

$$
\min_{f\in\Delta\mathcal{H}} \max_{\lambda\in\Lambda} \mathcal{L}_\mathcal{D}(f, \lambda) := -\mathbb{E}_{h\sim f}\mathbb{E}_\mathcal{D}\left[\sum_{\boldsymbol{a}\in\mathcal{A}} u(\boldsymbol{a}, y) \cdot b(h(x), \boldsymbol{a})\right]
$$

$$
+ \sum_{s\in\{+,-\}} \sum_{i=1}^N \sum_{j=1}^d \sum_{a_i\in A_i} \lambda_{s,i,j,a_i} s(\mathbb{E}_f\mathbb{E}_D[(h(x)_j - y_j) \cdot b_i(h(x), a_i)] - \gamma).
$$

$$(4)$$

We first show that an approximate equilibrium $(f, \lambda)$ to the $C$-bounded minimax game is indeed an approximately optimal solution to the original problem Eq. (2). We prove that $f$ achieves approximately optimal utility with respect to $\mathrm{OPT}(\mathcal{H}, \mathcal{D}, \gamma)$, while ensuring that its decision calibration error satisfies the $\gamma$-constraint up to an approximation error introduced by solving the $C$-bounded minimax game.

**Lemma 3.1.** *For an $\epsilon$-approximate equilibrium of the $C$-bounded minimax game $(f, \lambda)$. For the original unbounded constraint optimization problem Eq.* (2)*, we have that $\mathbb{E}_{h\sim f}\mathbb{E}_\mathcal{D}[u(\boldsymbol{a}, y) \cdot b(h(x), \boldsymbol{a})] \geq \mathrm{OPT}(\mathcal{H}, \mathcal{D}, \gamma) - 2\epsilon$, and $\mathrm{DecCE}(f) \leq \gamma + \frac{1+2\epsilon}{C}$.*

Therefore, we reduce the sender's constrained optimization problem Eq. (2) to solving the equilibrium of the above bounded minimax game.

**Solving Bounded Minimax Games** We now move on to solve the $C$-bounded minimax game. Note that the domain $\lambda$ can be viewed as a scaling of the probability simplex. A natural choice of algorithm for this domain is a variant of the Hedge algorithm (Freund & Schapire, 1997), which is originally designed for the simplex. For simplicity, we scale the Hedge algorithm by $C$ while still referring to it as Hedge. For computing the best response of the minimization player, we assume access to an empirical risk minimization (ERM) oracle that finds the best deterministic predictor given the current $\lambda$. We formally define the ERM oracle as follows:

**Definition 3.1** (ERM oracle). *Let the loss function be*

$$
\ell_\lambda(h, x, y) = -\sum_{\boldsymbol{a}\in\mathcal{A}} u(\boldsymbol{a}, y) \cdot b(h(x), \boldsymbol{a})
$$

$$
+ \sum_{s\in\{+,-\}} \sum_{i=1}^N \sum_{j=1}^d \sum_{a_i\in A_i} \lambda_{s,i,j,a_i} s((h(x)_j - y_j) \cdot b_i(h(x), a_i) - \gamma),
$$

*given a dataset with data points $D = \{(x_1, y_1), ..., (x_n, y_n)\}$, the ERM oracle finds the best predictor that minimizes the empirical average loss:* $\mathrm{ERM}(D, \lambda) = \arg\min_{h\in\mathcal{H}} \frac{1}{n}\sum_{i=1}^n \ell_\lambda(h, x_i, y_i)$.

The ERM oracle is commonly assumed in the learning theory literature and can often be implemented in practice using standard optimization methods. For example, when $\mathcal{H}$ is a class of neural networks, the ERM oracle can be approximated by running heuristic methods such as stochastic gradient descent (SGD) with a smoothed surrogate of the indicator $b(h(x), \boldsymbol{a})$ and $b_i(h(x), \boldsymbol{a})$.

We are now ready to present our algorithm `PerDecCal` in Algorithm 1. `PerDecCal` is efficient the ERM oracle is called only once per iteration, and all other operations are computationally polynomial in $N, m, d$. Therefore, overall `PerDecCal` is oracle-efficient, requiring $O(\log(Nmd)/\epsilon^4)$ calls to the ERM oracle.

---

**Algorithm 1** `PerDecCal` (Persuasive Decision Calibration)

---

**Input:** A set of samples $D$, ERM oracle $\text{ERM}(D, \lambda)$, dual bound $C$ and tolerance $\gamma$.

1: Initialize $\lambda_1 = \frac{C}{2Nmd}\mathbf{1}$.
2: **for** $t = 1, \cdots, T$ **do**
3:    Learner best responds to $\lambda_t$:
4:      Use the ERM oracle to compute $h_t = \text{ERM}(D, \lambda_t)$.
5:    Auditor runs Hedge to obtain $\lambda_{t+1}$:
6:      $\lambda_{t+1} = \text{Hedge}(c_{1:t})$ where $c_t(\lambda_{s,i,j,a_i}) = \lambda_{s,i,j,a_i} s(\mathbb{E}_D[(h_t(x)_j - y_j) \cdot b_i(h_t(x), a_i)] - \gamma)$.
7: **end for**
**Output:** $\hat{f} = \text{Uniform}(h_1, \cdots, h_T)$.

---

`PerDecCal` operates on the empirical dataset $D$ instead of the true distribution $\mathcal{D}$. Therefore, a finite-sample analysis is needed to show that an approximate equilibrium found for $\mathcal{L}_D(f, \lambda)$ also serves as an approximate equilibrium for $\mathcal{L}_{\mathcal{D}}(f, \lambda)$. We prove a uniform convergence result showing that the payoff $\mathcal{L}_{\mathcal{D}}(f, \lambda)$ can be uniformly approximated by $\mathcal{L}_D(f, \lambda)$.

**Lemma 3.2.** *We have* $|\mathcal{L}_{\mathcal{D}}(f, \lambda) - \mathcal{L}_D(f, \lambda)| \leq \sqrt{\frac{\ln \frac{|4\mathcal{H}|}{\delta}}{2n}} + C\sqrt{\frac{8\ln\frac{4|\mathcal{H}|Ndm}{\delta}}{n}}$ *for all* $f \in \Delta(\mathcal{H}), \lambda \in \Lambda$ *with probability* $1 - \delta$.

Then we can show that, with high probability, an approximate equilibrium under $\mathcal{L}_D(f, \lambda)$ is also an approximate equilibrium under $\mathcal{L}_{\mathcal{D}}(f, \lambda)$, completing the analysis of Theorem 3.1.

## 4 MATCHING THE BAYESIAN BENCHMARK

In this section, we show that the sender utility achieved by our algorithm matches that of a fully informed Bayesian sender who is restricted to send signals induced by the same class of decision-calibrated predictors. Specifically, we compare against a restricted Bayesian persuasion benchmark in which both the sender and the receiver have full knowledge of the distribution $\mathcal{D}$, and the sender is constrained to commit to signaling schemes induced by $\mathcal{H}$ (defined in Definition 4.2), rather than all possible signaling schemes. Since the classical Bayesian persuasion model inherently involves a single receiver, we focus on the comparison within the single-receiver setting.

**Bayesian Persuasion Benchmark** The distribution $\mathcal{D}$ over $\mathcal{X} \times \mathcal{Y}$ which induces a distribution $\mu_{\mathcal{D}}$ over means of the outcome $y$ conditional on the feature $x$: for any $\theta \in \mathcal{Y}$

$$\mu_{\mathcal{D}}(\theta) = \Pr_{(x,y)\sim\mathcal{D}}[\mathbb{E}[y \mid x] = \theta].$$

Here we use the fact that $\mathcal{Y}$ is convex. The set of state is $\Theta \subset \mathcal{Y}$ with prior $\mu_{\mathcal{D}}$. The receiver's action set $\mathcal{A}^{\text{BP}}$ equals to the action set $\mathcal{A}$ in the prediction setting with utility function $v^{\text{BP}} : \mathcal{A}^{\text{BP}} \times \Theta \to \mathbb{R}$. The sender has utility function $u^{\text{BP}} : \mathcal{A}^{\text{BP}} \times \Theta \to \mathbb{R}$. We have $v^{\text{BP}}(a, \theta) = \mathbb{E}_{y\sim\theta}[v(a, y)]$ and $u^{\text{BP}}(a, \theta) = \mathbb{E}_{y\sim\theta}[u(a, y)]$. Here we slightly abuse notation by writing $y \sim \theta$ to indicate that $y$ is drawn from a distribution with mean $\theta$. A signaling scheme $\pi : \Theta \to \Delta(S)$ that randomly maps states to a set $S$ of signals. Once the receiver observes a signal $s \in S$, they will update their belief from the prior $\mu$ to a posterior $\mu_s \in \Delta(\Theta)$ and consequently obtain a posterior mean that is in $\mathcal{Y}$. In other words, any signaling scheme will result in a distribution of posterior means $Q \in \Delta(\mathcal{Y})$.

We now establish the connection between decision calibration and the Bayesian persuasion benchmark introduced above. To do so, we use the notion of calibration as a bridge. Therefore, we begin by introduction the notion of calibration and presenting its relationship to Bayesian persuasion.

**Definition 4.1** (Calibration). *A randomized predictor* $f \in \Delta(\mathcal{H})$ *is said to be perfectly calibrated if*

$$\forall v \in \mathcal{Y}, \qquad \text{CE}(f) := \mathbb{E}_{h\sim f, (x,y)\sim\mathcal{D}}[(y - h(x))|h(x) = v] = 0.$$

The following lemma states that every signaling scheme corresponds to a perfectly calibrated predictor, in the sense that the distribution over posterior means induced by the signaling scheme coincides with the distribution over predictions induced by a randomized calibrated predictor, and vice versa.

**Lemma 4.1.** *Consider a randomized predictor $f \in \Delta(\mathcal{H}_{\mathrm{ALL}})$ where $\mathcal{H}_{\mathrm{ALL}} = \{h : \mathcal{X} \to \mathcal{Y}\}$ is the class of all possible deterministic predictors. There exists a distribution $Q_f \in \Delta(\mathcal{Y})$ such that for any $v \in \mathcal{Y}$, $Q_f(v) = \mathrm{Pr}_{h \sim f, (x,y) \sim \mathcal{D}}[h(x) = v]$. A distribution $Q \in \Delta(\mathcal{Y})$ corresponds to the distribution over posterior means induced by some signaling scheme if and only if it is the prediction distribution $Q_f$ of a perfectly calibrated predictor $f$.*

Note that the receiver's utility function $v_i$ in the prediction setting is linear in the outcome $y$ and the corresponding utility function $v^{\mathrm{BP}}$ in the Bayesian persuasion setting is linear in the conditional mean $\theta$. By Lemma 4.1, it follows immediately that for the receiver, best responding to the predictions of a perfectly calibrated predictor is equivalent to best responding to the posterior means induced by the corresponding signaling scheme. Therefore, the calibrated predictor and the corresponding signaling scheme lead to the same sender's utility.

Now we present the connection between decision calibration and calibration. The next lemma says that any decision-calibrated predictor can be converted to a calibrated predictor without decreasing the sender's expected utility. Zhao et al. (2021) made a similar observation, though their result applies only to deterministic predictors, whereas we extend the analysis to randomized predictors.

**Lemma 4.2.** *For any randomized predictor $f$ is perfectly decision calibrated, we can construct a randomized predictor $f'$ such that (i) $f'$ is perfectly calibrated; (ii) the sender obtains the same expected utility under $f$ and $f'$.*

We are now ready to define the set of signaling schemes induced by $\mathcal{H}$ that we consider in our Bayesian persuasion benchmark.

**Definition 4.2** (Signaling scheme class induced by $\mathcal{H}$). *Given any class of deterministic predictors $\mathcal{H}$,*

1. *We define $\mathcal{F}_{\mathrm{DCAL}}(\mathcal{H})$ as the class of randomized predictor over $\mathcal{H}$ that is perfectly decision calibrated.*

2. *For any $f \in \mathcal{F}_{\mathrm{DCAL}}(\mathcal{H})$, let $f'$ be the perfectly calibrated predictor constructed by Lemma 4.2. Define $\mathcal{F}_{\mathrm{CAL}}(\mathcal{H})$ as the class of all such predictors $f'$.*

3. *For any $f'$ in $\mathcal{F}_{\mathrm{CAL}}(\mathcal{H})$, let $\pi_{f'}$ be the corresponding signaling scheme by Lemma 4.1. Define $\Pi_{\mathcal{H}}$ be the class of all such signaling schemes $\pi_{f'}$. We say that $\Pi_{\mathcal{H}}$ is the class of signaling schemes induced by $\mathcal{H}$.*

Finally we are ready to present our main result in this section.

**Theorem 4.1.** *Given at least $O(\frac{\ln(|\mathcal{H}|dm/\delta)}{\epsilon^4})$ samples, with probability $1 - \delta$, `PerDecCal` can output a predictor $\hat{f}$ such that the expected sender's utility under $\hat{f}$ is no worse than $\mathrm{BayesOPT}(\mu_{\mathcal{D}}, \Pi_{\mathcal{H}}) - \epsilon$ where we denote the optimal sender utility under our Bayesian persuasion benchmark as $\mathrm{BayesOPT}(\mu_{\mathcal{D}}, \Pi_{\mathcal{H}})$.*

Theorem 4.1 says the sender utility achieved by our algorithm matches that of a fully informed Bayesian sender who is restricted to send signals induced by the same class of predictors $\mathcal{H}$. Note that Theorem 4.1 naturally extends to the multi-receiver setting. In this case, the predictor can still match the Bayesian benchmark, where each receiver updates their posterior based on their own recommended action as the signal. However, if the receivers instead take the recommended joint action across all receivers as the signal, the decision calibration constraints need be modified to ensure unbiasedness with respect to the joint indicator $b(h(x), \boldsymbol{a})$ instead of individual $b_i(h(x), a)$.

## 5 PERSUASIVE PREDICTION UNDER INFINITE HYPOTHESIS CLASS

In this section, we turn to the more general case where $|\mathcal{H}|$ can potentially be infinite. In this setting, strict best responses pose a challenge due to their discontinuity: even when two predictors $h_1, h_2 \in \mathcal{H}$ are close—i.e., $\sup_{x \in \mathcal{X}} \|h_1(x) - h_2(x)\|_{\infty}$ is small—the sender's utility under strict best response can differ by a constant. We illustrate this in the following example.

**Example 5.1** (Discontinuity from best response). *Consider a distribution $\mathcal{D}$ over $\mathcal{X} \times \mathcal{Y}$ such that $\mathcal{Y} = [0,1]$ and for any $x \in \mathcal{X}$, $\Pr[y \mid x] = 0.5$. There is a single receiver, i.e. $N = 1$, who has two actions $\mathcal{A} = \{a, a'\}$ to choose from. The receiver has a utility function $v : \mathcal{A} \times \mathcal{Y} \to [0,1]$ such that $v(a, y) = y, v(a', y) = 1 - y$. In other words, the receiver's best response is $a$ when $y \leq 0.5$ and is $a'$ when $y > 0.5$. The sender has a utility function $v : \mathcal{A} \times \mathcal{Y} \to [0,1]$ that only depends on the receiver's action: $u(a, y) = 1, u(a', y) = 0$. Now consider the following two predictors: for any $x \in \mathcal{X}$, $h_1(x) = 0.5 - \epsilon$ and $h_2(x) = 0.5 + \epsilon$. It is not hard to verify that both $h_1$ and $h_2$ are $\epsilon$-decision calibrated predictor under distribution $\mathcal{D}$. However, for any $\epsilon \in (0, 0.5)$, the sender's expected utility is 1 under $h_1$, but 0 under $h_2$.*

Because of the discontinuity of the best response, even if $\mathcal{H}$ has a bounded complexity measure—such as a covering number or Rademacher complexity—it can be difficult to estimate the sender's utility (which best depends on receivers' best responses) uniformly over all $h \in \mathcal{H}$ from data. To overcome this challenge, we consider a *smoothed* version of the best response decision rule, commonly known as the *quantal response* model in economics and decision theory. This model has been extensively studied in the literature (McFadden et al., 1976; McKelvey & Palfrey, 1995) as it captures more realistic receiver behavior in the presence of noise, uncertainty, or bounded rationality. Unlike strict best responses, it allows receivers to probabilistically favor better actions while still occasionally choosing suboptimal ones, providing a smoother and more practical behavior model.

**Definition 5.1** (Quantal Response). *For any $i \in [N]$, the $i$-th receiver with utility function $v_i$ responds to a prediction $h(x)$ according to the following $\eta$-quantal response:*

$$\tilde{b}_i(h(x), a_i) = \frac{e^{\eta v_i(a_i, h(x))}}{\sum_{a_i' \in \mathcal{A}_i} e^{\eta v_i(a_i', h(x))}}.$$

*$\tilde{b}_i(h(x), a_i)$ denotes the probability that receiver $i$ selects action $a_i$ given the prediction $h(x)$. Here, $\eta > 0$ is the inverse temperature parameter, where as $\eta \to +\infty$, the receiver's behavior approaches the strict best response.*

Analogously, we define a smoothed version of decision calibration when receivers follow quantal response model.

**Definition 5.2** (Smoothed Decision Calibration). *A randomized predictor $f \in \Delta(\mathcal{H})$ is said to be perfectly smoothed decision calibrated if*

$$\text{SmDecCE}(f) := \max_{i \in [N]} \max_{j \in [d]} \max_{a \in \mathcal{A}_i} \left| \mathbb{E}_{h \sim f} \mathbb{E}_{(x,y) \sim \mathcal{D}} \left[ (y_j - h(x)_j) \cdot \tilde{b}_i(h(x), a) \right] \right| = 0.$$

*Moreover, $f$ is said to be $\epsilon$-decision calibrated if $\text{SmDecCE}(f) \leq \epsilon$.*

It can be shown that receivers have no regret when following the quantal response to a smoothed calibrated predictor; we refer the reader to Appendix F for a detailed discussion. Similar to Section 3, we design an oracle-effcient algorithm `SmPerDecCal` (Algorithm 2) for persuasive prediction given quantal response and $|\mathcal{H}|$ can be infinite.

**Theorem 5.1.** *Suppose `SmPerDecCal` runs for $T = O(\log(Nmd)/\epsilon^4)$ rounds and is given a dataset $D$ drawn i.i.d of size $n \geq O(\ln \frac{\mathcal{N}(\mathcal{H}, d_\infty, \frac{\epsilon^2}{\eta L}) N d m}{\delta} / \epsilon^4)$. With probability at least $1 - \delta$, it outputs $\hat{f}$ that satisfies*

*1. $\text{SmDecCE}(\hat{f}) \leq \gamma + \epsilon$.*

*2. Suppose the receivers play $\eta$-quantal response to $\hat{f}$. Then the receivers obtain swap regret bounded by $2mL(\gamma + \epsilon) + \frac{\ln m + 1}{\eta}$. The sender achieves $\epsilon$-optimal utility:*

$$\mathbb{E}_{h \sim f} \mathbb{E}_{\mathcal{D}} \left[ u(\boldsymbol{a}, y) \cdot \tilde{b}(h(x), \boldsymbol{a}) \right] \geq \text{OPT}(\mathcal{H}, \mathcal{D}, \gamma) - \epsilon.$$

Theorem 5.1 shows that, with enough sample size, `SmPerDecCal` learns a predictor $\hat{f}$ that achieves nearly optimal utility compared to the best in-class $\gamma$-smoothed-decision-calibrated predictor, while ensuring that its smoothed decision calibration error exceeds $\gamma$ by at most $\epsilon$.

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

## A ETHICS STATEMENT

This paper develops a theoretical and algorithmic framework for persuasive prediction under decision calibration. Our contributions are methodological: we provide formal definitions, theorems, and algorithms, without using human subjects, sensitive personal data, or deployed systems. As such, we do not identify immediate ethical risks directly arising from this work.

Nevertheless, it is important to acknowledge that algorithms for persuasive prediction may have societal implications if applied in practice. In domains such as credit scoring, lending, or healthcare, predictive persuasion could create incentives for strategic manipulation. These potential issues fall outside the scope of our study, but we emphasize that our contribution is purely theoretical and algorithmic, and real-world deployment of related methods must be approached with care and oversight.

## B REPRODUCIBILITY STATEMENT

We have provided complete formal definitions, theorems, and proofs in the main text and appendices, ensuring that all results can be independently verified. Our algorithms are fully specified in pseudocode with stated assumptions and sample complexity guarantees. Since our contributions are theoretical, no external datasets are required.

## C USE OF LARGE LANGUAGE MODELS

We used large language models only as writing assistants to improve readability and polish the presentation.

## D ADDITIONAL RELATED WORK

Our work is related to a growing line of work calibration in decision-making settings. The seminal work of Foster & Vohra (1999) showed that a decision maker who best responds to calibrated

forecasts obtain diminishing internal regret. More recent studies extend this result by proposing refined notions of calibration that are more efficient to achieve and offer fine-grained regret guarantees (Zhao et al., 2021; Kleinberg et al., 2023; Hu & Wu, 2024; Roth & Shi, 2024; Fishelson et al., 2025; Luo et al., 2025; Tang et al., 2025). Building on (multi)calibration, Gopalan et al. (2021) introduced the notion of omniprediction, which aims to construct a single predictor that guarantees no worse loss than a family of predetermined benchmarks for all the downstream receivers in a class, followed by Gopalan et al. (2022; 2024); Garg et al. (2024); Dwork et al. (2024); Okoroafor et al. (2025); Lu et al. (2025). In contrast to these works, we not only aim to achieve a specific notion of calibration (decision calibration, in our case), but also seek to approximately maximize the sender's utility among all such calibrated predictors.

Our work shares the common goal of replacing prior knowledge with data, aligning with many works in mechanism design, such as auction design (Balcan et al., 2008; Cole & Roughgarden, 2014; Morgenstern & Roughgarden, 2015; Daskalakis & Syrgkanis, 2016; Syrgkanis, 2017; Dudík et al., 2020; Fu & Lin, 2020), Stackelberg game (Balcan et al., 2015; Camara et al., 2020; Collina et al., 2024), algorithm discrimination (Cummings et al., 2020) and recommendation system (Immorlica et al., 2018).

More broadly, a growing body of work in economics aims to relax the assumption of perfect prior knowledge rather than replace it entirely, such as relaxing the prior to some kind of approximate agreement on the distribution (Artemov et al., 2013; Ollár & Penta, 2017) and robustness to prior distribution (Dworczak & Pavan, 2022; Kosterina, 2022). In contrast to these works, we adopt a data-driven approach to address the challenge of an unknown prior distribution. Parakhonyak & Sobolev (2025) also studies persuasion without a prior. Unlike our setting, which assumes sample access to the data distribution, they design a signaling scheme that minimizes the worst-case gap relative to the Bayesian benchmark.

From a technical perspective, the problem of solving constrained optimization through No Regret versus Best Response dynamics has been studied in the algorithmic fairness literature (Agarwal et al., 2019; 2018; Kearns et al., 2018; Globus-Harris et al., 2023).

## E  GLOBAL OPTIMALITY FOR FINITE-SIZE $\mathcal{X}$

In this section, we consider the case that $|\mathcal{X}| < \infty$, $\mathcal{Y} = \{0, 1\}$ and there is one receiver, i.e. $N = 1$. We show that it is sufficient to consider predictions in an instance-dependent discretization set. Fix any receiver's utility $v$, we slightly abuse notation, let $\mathcal{A} = \{a_1, \cdots, a_m\}$ denote the receiver's action set. Let $J_i = \{p \in [0,1] : a_i = \arg\max \mathbb{E}_{y \sim Ber(p)}[v(a, y)]\}$ for any $i \in [m]$. If there are ties, the receiver breaks ties in favor of the sender. Then we know that $J_i$ is an interval on [0,1] for any $i \in [m]$ and $\{J_i\}_{i=1}^m$ is a partition of $[0, 1]$. Let $\mathcal{Z}$ be the set of thresholds of adjacent best response intervals. We know $|\mathcal{Z}| \leq m - 1$. Let $\Theta = \{\Pr[y = 1 \mid x] : \forall x \in \mathcal{X}\}$. We know $|\Theta| \leq |\mathcal{X}|$.

**Definition E.1** (Instance-dependent discretization)**.** *For any positive $\epsilon < \min_{i \in m} \text{len}(J_i)$, define discretized set $S_\epsilon$ of space [0,1] as*

$$S_\epsilon \triangleq (\{0, \epsilon, 2\epsilon, \cdots\} \cap [0,1]) \cup \mathcal{Z} \cup \Theta.$$

**Definition E.2** (Discretized predictor set)**.** *Let $\mathcal{H}_\epsilon = \{h : \mathcal{X} \to S_\epsilon\}$ be the set of all possible predictors whose predictions are always in the instance-dependent discretization set $S_\epsilon$.*

Note that $\mathcal{H}_\epsilon$ is finite with size $|S_\epsilon|^{\mathcal{X}}$. The following theorem shows that any randomized predictor in $\Delta(\mathcal{H}_{\text{ALL}})$ can be converted to a randomized predictor in $\Delta(\mathcal{H}_\epsilon)$ without changing the sender's utility and increase the decision calibration error up to $\epsilon$.

**Theorem E.1.** *For any $\epsilon > 0$, for any randomized predictor $f \in \Delta(\mathcal{H}_{\text{ALL}})$ that is $\gamma$-decision calibrated, we can construct a randomized predictor $f' \in \Delta(\mathcal{H}_\epsilon)$ such that (1) the sender obtains the same expected utility under $f$ and $f'$ (2) $f'$ is $(\gamma + \epsilon)$-decision calibrated.*

*Proof.* Define the range of $f$ as $\text{range}(f) = \bigcup_{h \in \text{supp} f} \text{range}(h)$. Consider the following rounding function $r : \text{range}(f) \to S_\epsilon$:

$$r(v) = p \quad v \in J_i, \; p = \arg \inf_{p' \in (S_\epsilon \cap J_i)} |p' - v|.$$

For any $h \sim f \in \Delta(\mathcal{H}_{\mathrm{ALL}})$, we construct $h' \in \mathcal{H}_\epsilon$ as $h' : x \mapsto r(h(x))$. Let $f'$ be the distribution over such $h'$. By the definition of $S_\epsilon$, we have that (1) $|v - r(v)| < \epsilon$ (2) the receiver has the same best response under $v$ and $r(v)$. Therefore, the sender obtains the same expected utility under $f$ and $f'$. And we have

$$
\begin{aligned}
\left| \mathbb{E}_{h' \sim f'} \mathbb{E}_{(x,y) \sim \mathcal{D}} [(y - h(x)) \cdot b(h'(x), a)] \right| &= \left| \mathbb{E}_{h' \sim f'} \mathbb{E}_{(x,y) \sim \mathcal{D}} [(y - h(x)) \cdot b(h(x), a)] \right| \\
&\leq \left| \mathbb{E}_{h' \sim f'} \mathbb{E}_{(x,y) \sim \mathcal{D}} [(y - h(x)) \cdot b(h(x), a)] \right| \\
&\quad + \left| \mathbb{E}_{h' \sim f'} \mathbb{E}_{(x,y) \sim \mathcal{D}} [(h'(x) - h(x)) \cdot b(h(x), a)] \right| \\
&= \left| \mathbb{E}_{h \sim f} \mathbb{E}_{(x,y) \sim \mathcal{D}} [(y - h(x)) \cdot b(h(x), a)] \right| \\
&\quad + \left| \mathbb{E}_{h \sim f} \mathbb{E}_{(x,y) \sim \mathcal{D}} [(h'(x) - h(x)) \cdot b(h(x), a)] \right| \\
&\leq \gamma + \epsilon.
\end{aligned}
$$

$\square$

This theorem implies that, at least in the special case considered in this section, to find the optimal decision-calibrated predictor randomized over all deterministic predictors, it suffices to consider a finite subset of deterministic predictors $\mathcal{H}_\epsilon$.

# F   NO REGRET GUARANTEES OF DECISION CALIBRATION

## F.1   NO REGRET GUARANTEES OF STRICT DECISION CALIBRATION

We first prove that approximate decision calibrated predictor gives the downstream agent no swap regret best responding to it. Noarov et al. (2023) proved it for deterministic predictor in the online calibration setting. We provide our proof for randomized predictor in the batch settting here for completeness.

**Theorem 2.1** (No Swap Regret via Decision Calibration). *If a predictor $f$ is $\epsilon$-decision calibrated, then it has at most $2L|A|\epsilon$-swap regret.*

*Proof.* We prove the result for any receiver $i \in [N]$.

$$
\mathbb{E}_{h \sim f} \mathbb{E}_{\mathcal{D}} \left[ \sum_a v_i(\phi(a), y) \cdot b_i(h(x), a) \right] - \mathbb{E}_{h \sim f} \mathbb{E}_{\mathcal{D}} \left[ \sum_a v_i(a, y) \cdot b_i(h(x), a) \right]
$$

$$
= \mathbb{E}_{h \sim f} \mathbb{E}_{\mathcal{D}} \left[ \sum_a v_i(\phi(a), y) \cdot b_i(h(x), a) \right] - \mathbb{E}_{h \sim f} \mathbb{E}_{\mathcal{D}} \left[ \sum_a v_i(\phi(a), h(x)) \cdot b_i(h(x), a) \right]
$$

$$
+ \mathbb{E}_{h \sim f} \mathbb{E}_{\mathcal{D}} \left[ \sum_a v_i(\phi(a), h(x)) \cdot b_i(h(x), a) \right] - \mathbb{E}_{h \sim f} \mathbb{E}_{\mathcal{D}} \left[ \sum_a v_i(a, h(x)) \cdot b_i(h(x), a) \right]
$$

$$
+ \mathbb{E}_{h \sim f} \mathbb{E}_{\mathcal{D}} \left[ \sum_a v_i(a, h(x)) \cdot b_i(h(x), a) \right] - \mathbb{E}_{h \sim f} \mathbb{E}_{\mathcal{D}} \left[ \sum_a v_i(a, y) \cdot b_i(h(x), a) \right]
$$

When $f$ is $\epsilon$-decision-calibrated, we know that

$$
\mathbb{E}_{h \sim f} \mathbb{E}_{\mathcal{D}} \left[ \sum_a v_i(\phi(a), y) \cdot b_i(h(x), a) \right] - \mathbb{E}_{h \sim f} \mathbb{E}_{\mathcal{D}} \left[ \sum_a v_i(\phi(a), h(x)) \cdot b_i(h(x), a) \right]
$$

$$
= \sum_a \mathbb{E}_{h \sim f} \mathbb{E}_{\mathcal{D}} \left[ \sum_a v_i(\phi(a), y - h(x)) \cdot b_i(h(x), a) \right]
$$

$$
\leq \sum_a L\epsilon = L|A|\epsilon.
$$

Similarly we can prove that

$$
\mathbb{E}_{h \sim f} \mathbb{E}_{\mathcal{D}} \left[ \sum_a v_i(a, h(x)) \cdot b_i(h(x), a) \right] - \mathbb{E}_{h \sim f} \mathbb{E}_{\mathcal{D}} \left[ \sum_a v_i(a, y) \cdot b_i(h(x), a) \right] \leq L|A|\epsilon.
$$

Since $b_i(h(x), a)$ plays the best response given the prediction $h(x)$, we know that

$$\mathbb{E}_{h \sim f} \mathbb{E}_{\mathcal{D}} \left[ \sum_a v_i(\phi(a), h(x)) \cdot b_i(h(x), a) \right] - \mathbb{E}_{h \sim f} \mathbb{E}_{\mathcal{D}} \left[ \sum_a v_i(a, h(x)) \cdot b_i(h(x), a) \right] \le 0.$$

Putting them together, we have

$$\mathbb{E}_{h \sim f} \mathbb{E}_{\mathcal{D}} \left[ \sum_a v_i(\phi(a), y) \cdot b_i(h(x), a) \right] - \mathbb{E}_{h \sim f} \mathbb{E}_{\mathcal{D}} \left[ \sum_a v_i(a, y) \cdot b_i(h(x), a) \right] \le 2L|A|\epsilon.$$

$\square$

**Definition F.1** (Type Regret). *We say that a predictor $f$ achieves $\epsilon$-type regret if, for any receiver $i, i' \in [N], \phi : A \to A,$*

$$\mathbb{E}_{h \sim f} \mathbb{E}_{\mathcal{D}} \left[ \sum_a v_i(a, y) \cdot b_{i'}(h(x), a) \right] \le \mathbb{E}_{h \sim f} \mathbb{E}_{\mathcal{D}} \left[ \sum_a v_i(a, y) \cdot b_i(h(x), a) \right] + \epsilon.$$

Now we introduce a different notion of regret, named *type regret*. Type regret is first introduced by Zhao et al. (2021). Intuitively, it says that once the predictor gets decision calibrated with respect to a class of utility functions of receivers, the receivers will have no regret best responding according to another receiver's utility function instead their own. Zhao et al. (2021) proved that decision calibrated predictor achieves no type regret for the receivers. Here we state and prove the result for randomized predictors.

**Theorem F.1** (No Type Regret via Decision Calibration). *If a predictor $f$ is $\epsilon$-decision calibrated, then it satisfies $2L|A|\epsilon$-type regret.*

*Proof.* We prove the result for any receiver $i, i' \in [N]$.

$$\mathbb{E}_{h \sim f} \mathbb{E}_{\mathcal{D}} \left[ \sum_a v_i(a, y) \cdot b_{i'}(h(x), a) \right] - \mathbb{E}_{h \sim f} \mathbb{E}_{\mathcal{D}} \left[ \sum_a v_i(a, y) \cdot b_i(h(x), a) \right]$$

$$= \mathbb{E}_{h \sim f} \mathbb{E}_{\mathcal{D}} \left[ \sum_a v_i(a, y) \cdot b_{i'}(h(x), a) \right] - \mathbb{E}_{h \sim f} \mathbb{E}_{\mathcal{D}} \left[ \sum_a v_i(a, h(x)) \cdot b_{i'}(h(x), a) \right]$$

$$+ \mathbb{E}_{h \sim f} \mathbb{E}_{\mathcal{D}} \left[ \sum_a v_i(a, h(x)) \cdot b_{i'}(h(x), a) \right] - \mathbb{E}_{h \sim f} \mathbb{E}_{\mathcal{D}} \left[ \sum_a v_i(a, h(x)) \cdot b_i(h(x), a) \right]$$

$$+ \mathbb{E}_{h \sim f} \mathbb{E}_{\mathcal{D}} \left[ \sum_a v_i(a, h(x)) \cdot b_i(h(x), a) \right] - \mathbb{E}_{h \sim f} \mathbb{E}_{\mathcal{D}} \left[ \sum_a v_i(a, y) \cdot b_i(h(x), a) \right]$$

When $f$ is $\epsilon$-decision-calibrated, we know that

$$\mathbb{E}_{h \sim f} \mathbb{E}_{\mathcal{D}} \left[ \sum_a v_i(a, y) \cdot b_{i'}(h(x), a) \right] - \mathbb{E}_{h \sim f} \mathbb{E}_{\mathcal{D}} \left[ \sum_a v_i(a, h(x)) \cdot b_{i'}(h(x), a) \right]$$

$$= \sum_a \mathbb{E}_{h \sim f} \mathbb{E}_{\mathcal{D}} \left[ \sum_a v_i(a, y - h(x)) \cdot b_{i'}(h(x), a) \right]$$

$$\le \sum_a L\epsilon = L|A|\epsilon.$$

Similarly we can prove that

$$\mathbb{E}_{h \sim f} \mathbb{E}_{\mathcal{D}} \left[ \sum_a v_i(a, h(x)) \cdot b_{i'}(h(x), a) \right] - \mathbb{E}_{h \sim f} \mathbb{E}_{\mathcal{D}} \left[ \sum_a v_i(a, h(x)) \cdot b_i(h(x), a) \right] \le L|A|\epsilon.$$

Since $b_i(h(x), a)$ plays the best response given the prediction $h(x)$, we know that

$$\mathbb{E}_{h \sim f} \mathbb{E}_{\mathcal{D}} \left[ \sum_a v_i(a, h(x)) \cdot b_{i'}(h(x), a) \right] - \mathbb{E}_{h \sim f} \mathbb{E}_{\mathcal{D}} \left[ \sum_a v_i(a, h(x)) \cdot b_i(h(x), a) \right] \le 0.$$

Putting them together, we have

$$\mathbb{E}_{h \sim f} \mathbb{E}_{\mathcal{D}} \left[ \sum_a v_i(a, y) \cdot b_{i'}(h(x), a) \right] - \mathbb{E}_{h \sim f} \mathbb{E}_{\mathcal{D}} \left[ \sum_a v_i(a, y) \cdot b_i(h(x), a) \right] \leq 2L|A|\epsilon.$$

$\square$

We also introduce a new notion of regret which we call swap-type-regret, which intuitively capture the case where the receivers can first pretend that they were another receiver and then swap the corresponding best-response action. We formally define it as follows:

**Definition F.2** (Swap-Type Regret). *We say that a predictor $f$ achieves $\epsilon$-swap-type regret if, for any receiver $i, i' \in [N]$, mapping function $\phi : A \to A$,*

$$\mathbb{E}_{h \sim f} \mathbb{E}_{\mathcal{D}} \left[ \sum_a v_i(\phi(a), y) \cdot b_{i'}(h(x), a) \right] \leq \mathbb{E}_{h \sim f} \mathbb{E}_{\mathcal{D}} \left[ \sum_a v_i(a, y) \cdot b_i(h(x), a) \right] + \epsilon.$$

**Theorem F.2** (No Swap-Type Regret via Decision Calibration). *If a predictor $f$ is $\epsilon$-decision calibrated, then it satisfies $2L|A|\epsilon$-swap-type regret.*

*Proof.* The proof is similar to the proofs of Theorem 2.1 and Theorem F.1. We can similarly prove that

$$\mathbb{E}_{h \sim f} \mathbb{E}_{\mathcal{D}} \left[ \sum_a v_i(\phi(a), y) \cdot b_{i'}(h(x), a) \right] - \mathbb{E}_{h \sim f} \mathbb{E}_{\mathcal{D}} \left[ \sum_a v_i(\phi(a), h(x)) \cdot b_{i'}(h(x), a) \right] \leq L|A|\epsilon,$$

and

$$\mathbb{E}_{h \sim f} \mathbb{E}_{\mathcal{D}} \left[ \sum_a v_i(a, h(x)) \cdot b_i(h(x), a) \right] - \mathbb{E}_{h \sim f} \mathbb{E}_{\mathcal{D}} \left[ \sum_a v_i(a, y) \cdot b_i(h(x), a) \right] \leq L|A|\epsilon.$$

From the fact that $b_i(h(x), a)$ selects the best response action, we also have

$$\mathbb{E}_{h \sim f} \mathbb{E}_{\mathcal{D}} \left[ \sum_a v_i(\phi(a), h(x)) \cdot b_{i'}(h(x), a) \right] - \mathbb{E}_{h \sim f} \mathbb{E}_{\mathcal{D}} \left[ \sum_a v_i(a, h(x)) \cdot b_i(h(x), a) \right] \leq 0.$$

Putting them together completes the proof. $\square$

F.2 NO REGRET GUARANTEES OF SMOOTHED DECISION CALIBRATION

We provide analogous result for the behavior model where the receivers follow quantal response. We first define the three notions of regret for quantal response.

**Definition F.3** (Swap Regret under Quantal Response). *We say that a predictor $f$ achieves $\epsilon$-swap regret for receivers that follow quantal response rule if, for any receiver $i \in [N]$, mapping function $\phi : A \to A$,*

$$\mathbb{E}_{h \sim f} \mathbb{E}_{\mathcal{D}} \left[ \sum_a v_i(\phi(a), y) \cdot \tilde{b}_i(h(x), a) \right] \leq \mathbb{E}_{h \sim f} \mathbb{E}_{\mathcal{D}} \left[ \sum_a v_i(a, y) \cdot \tilde{b}_i(h(x), a) \right] + \epsilon.$$

**Definition F.4** (Type Regret under Quantal Response). *We say that a predictor $f$ achieves $\epsilon$-type regret for receivers that follow quantal response rule if, for any receiver $i, i' \in [N], \phi : A \to A$,*

$$\mathbb{E}_{h \sim f} \mathbb{E}_{\mathcal{D}} \left[ \sum_a v_i(a, y) \cdot \tilde{b}_{i'}(h(x), a) \right] \leq \mathbb{E}_{h \sim f} \mathbb{E}_{\mathcal{D}} \left[ \sum_a v_i(a, y) \cdot \tilde{b}_i(h(x), a) \right] + \epsilon.$$

**Definition F.5** (Swap-Type under Quantal Response). *We say that a predictor $f$ achieves $\epsilon$-swap-type regret for receivers that follow quantal response rule if, for any receiver $i, i' \in [N]$, mapping function $\phi : A \to A$,*

$$\mathbb{E}_{h \sim f} \mathbb{E}_{\mathcal{D}} \left[ \sum_a v_i(\phi(a), y) \cdot \tilde{b}_{i'}(h(x), a) \right] \leq \mathbb{E}_{h \sim f} \mathbb{E}_{\mathcal{D}} \left[ \sum_a v_i(a, y) \cdot \tilde{b}_i(h(x), a) \right] + \epsilon.$$

We now present the analogous no regret guarantee for quantal response receivers. Swap regret for quantal response receivers are discussed in Roth & Shi (2024) in the online setting for deterministic predictors. Here, we provide the result for randomized predictors in the batch setting. Type regret for quantal response receivers are discussed in Tang et al. (2025) for deterministic predictors. Here, we state and prove the result for randomized predictors.

**Theorem F.3.** *If a predictor $f$ is $\epsilon$-smoothed-decision calibrated, then it satisfies $2L|A|\epsilon + \frac{\ln|A|+1}{\eta}$-swap/type/swap-type regret.*

To prove Theorem F.3, we first present a lemma proved by Roth & Shi (2024), which shows that the utility that a receiver gets when they play quantal response with respect to the true outcome will be close that when they strictly best respond.

**Lemma F.1** (Roth & Shi (2024)). *For any utility function $v$ and $y \in \mathcal{Y}$, let $a^* = \arg\max_a v(a, y)$, we have*

$$\sum_a v(a, y)\tilde{b}(y, a) \geq v(a^*, y) - \frac{\ln|A|+1}{\eta}.$$

*Proof of Theorem F.3.* We only provide proof for swap-type regret as it is the strongest. The guarantees for swap regret and type regret are simple corollaries by considering $\phi$ to be the identical mapping and $i = i'$.

For any $\phi : \mathcal{A} \times \mathcal{A}, i, i' \in [N]$, we have

$$\mathbb{E}_{h\sim f}\mathbb{E}_{\mathcal{D}}\left[\sum_a v_i(\phi(a), y) \cdot \tilde{b}_{i'}(h(x), a)\right] - \mathbb{E}_{h\sim f}\mathbb{E}_{\mathcal{D}}\left[\sum_a v_i(a, y) \cdot \tilde{b}_i(h(x), a)\right]$$

$$\leq \mathbb{E}_{h\sim f}\mathbb{E}_{\mathcal{D}}\left[\sum_a v_i(\phi(a), y) \cdot \tilde{b}_{i'}(h(x), a)\right] - \mathbb{E}_{h\sim f}\mathbb{E}_{\mathcal{D}}\left[\sum_a v_i(\phi(a), h(x)) \cdot \tilde{b}_{i'}(h(x), a)\right]$$

$$+ \mathbb{E}_{h\sim f}\mathbb{E}_{\mathcal{D}}\left[\sum_a v_i(\phi(a), h(x)) \cdot \tilde{b}_{i'}(h(x), a)\right] - \mathbb{E}_{h\sim f}\mathbb{E}_{\mathcal{D}}\left[\sum_a v_i(a, h(x)) \cdot \tilde{b}_i(h(x), a)\right]$$

$$+ \mathbb{E}_{h\sim f}\mathbb{E}_{\mathcal{D}}\left[\sum_a v_i(a, h(x)) \cdot \tilde{b}_i(h(x), a)\right] - \mathbb{E}_{h\sim f}\mathbb{E}_{\mathcal{D}}\left[\sum_a v_i(a, y) \cdot \tilde{b}_i(h(x), a)\right]$$

From the definition of decision calibration, we know that

$$\mathbb{E}_{h\sim f}\mathbb{E}_{\mathcal{D}}\left[\sum_a v_i(\phi(a), y) \cdot \tilde{b}_{i'}(h(x), a)\right] - \mathbb{E}_{h\sim f}\mathbb{E}_{\mathcal{D}}\left[\sum_a v_i(\phi(a), h(x)) \cdot \tilde{b}_{i'}(h(x), a)\right] \leq L|A|\epsilon$$

and

$$\mathbb{E}_{h\sim f}\mathbb{E}_{\mathcal{D}}\left[\sum_a v_i(a, h(x)) \cdot \tilde{b}_i(h(x), a)\right] - \mathbb{E}_{h\sim f}\mathbb{E}_{\mathcal{D}}\left[\sum_a v_i(a, y) \cdot \tilde{b}_i(h(x), a)\right] \leq L|A|\epsilon.$$

By Lemma F.1, we know that

$$\mathbb{E}_{h\sim f}\mathbb{E}_{\mathcal{D}}\left[\sum_a v_i(a, h(x)) \cdot \tilde{b}_i(h(x), a)\right] \geq \mathbb{E}_{h\sim f}\mathbb{E}_{\mathcal{D}}\left[\sum_a v_i(a, h(x)) \cdot b_i(h(x), a)\right] - \frac{\ln|A|+1}{\eta}.$$

Therefore, we have

$$\mathbb{E}_{h\sim f}\mathbb{E}_{\mathcal{D}}\left[\sum_a v_i(\phi(a), h(x)) \cdot \tilde{b}_{i'}(, h(x), a)\right] - \mathbb{E}_{h\sim f}\mathbb{E}_{\mathcal{D}}\left[\sum_a v_i(a, h(x)) \cdot \tilde{b}_i(h(x), a)\right]$$

$$\leq \mathbb{E}_{h\sim f}\mathbb{E}_{\mathcal{D}}\left[\sum_a v_i(\phi(a), h(x)) \cdot b_{i'}(, h(x), a)\right]$$

$$- \mathbb{E}_{h\sim f}\mathbb{E}_{\mathcal{D}}\left[\sum_a v_i(a, h(x)) \cdot b_i(, h(x), a)\right] + \frac{\ln|A|+1}{\eta}.$$

Since given $h(x)$, $b_i(h(x), a)$ is the optimal decision rule for $v_i$, we know that

$$\mathbb{E}_{h \sim f} \mathbb{E}_{\mathcal{D}} \left[ \sum_a v_i(\phi(a), h(x)) \cdot b_{i'}(h(x), a) \right] - \mathbb{E}_{h \sim f} \mathbb{E}_{\mathcal{D}} \left[ \sum_a v_i(a, h(x)) \cdot b_i(h(x), a) \right] \le 0.$$

Putting them together, we have that

$$\mathbb{E}_{h \sim f} \mathbb{E}_{\mathcal{D}} \left[ \sum_a v_i(\phi(a), y) \cdot b_{i'}(h(x), a) \right] - \mathbb{E}_{h \sim f} \mathbb{E}_{\mathcal{D}} \left[ \sum_a v_i(a, y) \cdot b_i(h(x), a) \right]$$

$$\le 2L|A|\epsilon + \frac{\ln |A| + 1}{\eta}.$$

$\square$

## G  STATISTICAL HARDNESS OF LEARNING AN OPTIMAL PREDICTOR WITH THE EXACT CONSTRAINTS

In this section, we show that it is statistically hard to learn the optimal predictor in the class of $\gamma$-decision calibrated predictor. Specially, we focus on $\gamma = 0$. Then an algorithm is called $(\varepsilon, \delta)$-bicriteria optimal if, there exist a function $n_0 : (0,1)^2 \to \mathbb{N}$ such that given $n \ge n_0(\epsilon, \delta)$ i.i.d. samples from an unknown distribution $\mathcal{D}$, it outputs a (possibly randomized) predictor $f$ such that, with probability at least $1 - \delta$,

(i) $\text{DecCE}(f) \le \varepsilon$    and    (ii) $\mathbb{E}_{h \sim f} \mathbb{E}_{(x,y) \sim \mathcal{D}} \left[ \sum_{a \in \mathcal{A}} u(a, y) \, b(h(x), a) \right] \ge \text{OPT}(\mathcal{H}, \mathcal{D}, \gamma) - \varepsilon.$

**Instance.**  Consider one receiver ($N$=1), outcomes $y \in \{0, 1\}$, a feature space $\mathcal{X} = \{0\}$, and a hypothesis class

$$\mathcal{H} = \{h_1, h_2\}, \qquad h_1(x) \equiv v_1, \ h_2(x) \equiv v_2, \quad 0 < v_1 < v_2 < 1.$$

Suppose the receiver's optimal action is $a_1$ if $y \ge v_1$ and $a_2$ otherwise. The sender receives utility 1 if the receiver chooses $a_1$, and 0 otherwise. Let $\mathcal{D}_-$ be the distribution with $x \equiv 0$ and $y \sim \text{Bern}(v_1)$, and $\mathcal{D}_+$ the same with $y \sim \text{Bern}(v_2)$. Define $\Delta := v_2 - v_1 = \text{TVD}(\mathcal{D}_-, \mathcal{D}_+)$ and assume $v_1, v_2 \in [\tau, 1 - \tau]$ for some $\tau \in (0, \frac{1}{2})$. Choose $\varepsilon$ so that $0 < \varepsilon < \Delta/2$.

For any (possibly randomized) $f \in \Delta(\mathcal{H})$,

$$\text{DecCE}(f) = \Big| \mathbb{E}_{h \sim f} \big[ \mathbb{E}[y] - h(0) \big] \Big|. \tag{5}$$

**Theorem G.1.** *For the instance above, any $(\varepsilon, \delta)$-bicriteria algorithm with $0 < \varepsilon < \Delta/2$ and $0 < \delta < \frac{1}{2}$ that succeeds using $n$ samples must satisfy*

$$n \ge \frac{2\,\tau(1 - \tau)\,(1 - 2\delta)^2}{\Delta^2}.$$

*Equivalently, the sample complexity is $\Omega((1 - 2\delta)^2 / \text{TVD}(\mathcal{D}_-, \mathcal{D}_+)^2)$.*

*Proof.* From equation 5, under $\mathcal{D}_-$ the feasibility condition $\text{DecCE}(f) \le \varepsilon$ implies $\mathbb{E}_{h \sim f}[h(0)] \in [v_1 - \varepsilon, v_1 + \varepsilon]$; under $\mathcal{D}_+$ it implies $\mathbb{E}_{h \sim f}[h(0)] \in [v_2 - \varepsilon, v_2 + \varepsilon]$. These intervals are disjoint because $2\varepsilon < \Delta$.

Under $\mathcal{D}_-$ (resp. $\mathcal{D}_+$), the exactly calibrated point mass $\delta_{h_1}$ (resp. $\delta_{h_2}$) is optimal, i.e., $\text{OPT}(\mathcal{H}, \mathcal{D}_\pm, 0) = 1$. Moreover, any $f \in \Delta(\mathcal{H})$ with $\text{DecCE}(f) \le \varepsilon$ is also $\varepsilon$-optimal for the objective.

Let an algorithm be $(\varepsilon, \delta)$-bicriteria optimal. With probability at least $1 - \delta$ it outputs an $f$ that is $\varepsilon$-feasible (and therefore $\varepsilon$-optimal). Consequently, the statistic $\mu := \mathbb{E}_{h \sim f}[h(0)]$ lies in disjoint intervals depending on whether the data come from $\mathcal{D}_-$ or $\mathcal{D}_+$. Thus declaring $\mathcal{D}_+$ iff $\mu > (v_1 + v_2)/2$ identifies the generating distribution with error at most $\delta$.

For any rule that distinguishes $\mathcal{D}_{-}^{\otimes n}$ from $\mathcal{D}_{+}^{\otimes n}$ with error at most $\delta$ one must have $\mathrm{TVD}(\mathcal{D}_{-}^{\otimes n}, \mathcal{D}_{+}^{\otimes n}) \geq 1 - 2\delta$. Pinsker's inequality and the KL chain rule give

$$\mathrm{TVD}(\mathcal{D}_{-}^{\otimes n}, \mathcal{D}_{+}^{\otimes n}) \leq \sqrt{\frac{n}{2} D_{\mathrm{KL}}(\mathcal{D}_{-} \,\|\, \mathcal{D}_{+})}.$$

For Bernoulli parameters $p, q \in [\tau, 1 - \tau]$, $D_{\mathrm{KL}}(\mathrm{Bern}(p) \,\|\, \mathrm{Bern}(q)) \leq (p - q)^2 / (\tau(1 - \tau))$. Here $|p - q| = \Delta$, so

$$\mathrm{TVD}(\mathcal{D}_{-}^{\otimes n}, \mathcal{D}_{+}^{\otimes n}) \leq \sqrt{\frac{n \, \Delta^2}{2 \, \tau(1 - \tau)}}.$$

Combining with $\mathrm{TVD} \geq 1 - 2\delta$ yields $\sqrt{n \, \Delta^2 / (2 \, \tau(1 - \tau))} \geq 1 - 2\delta$, i.e., $n \geq 2 \, \tau (1 - \tau) \, (1 - 2\delta)^2 / \Delta^2$.

$\square$

With zero slack ($\epsilon = 0$) the condition $\epsilon < \Delta/2$ holds for any $\Delta > 0$; since we can choose $\Delta = v_2 - v_1$ arbitrarily small while keeping $v_1, v_2 \in [\tau, 1 - \tau]$, Theorem G.1 gives $n \geq 2\tau(1 - \tau)(1 - 2\delta)^2 / \Delta^2 \to \infty$ as $\Delta \to 0$, i.e., the sample complexity is unbounded.

# H    MISSING PROOFS IN SECTION 3

We first state the lemma from Freund & Schapire (1996) which proves that when both players have low regret, their average play converge to an approximate equilibrium.

**Lemma H.1** (Freund & Schapire (1996)). *Consider a two-player zero-sum game where the min player chooses strategies from $\mathcal{P}$ and the max player chooses strategies from $\mathcal{Q}$. Assume $\mathcal{P}$ and $\mathcal{Q}$ are convex, and the utility function is bilinear in the players' strategies. If the sequence of plays satisfies sublinear regret for both players, i.e.,*

$$\min_{p \in \mathcal{P}} \sum_{t=1}^{T} u(p_t, q_t) - u(p, q_t) \leq \gamma_{\mathcal{P}} T, \quad and \quad \max_{q \in \mathcal{Q}} \sum_{t=1}^{T} u(p_t, q) - u(p_t, q_t) \leq \gamma_{\mathcal{Q}} T,$$

*then letting $\bar{p} = \frac{1}{T} \sum_{t=1}^{T} p_t$ and $\bar{q} = \frac{1}{T} \sum_{t=1}^{T} q_t$, we have that $(\bar{p}, \bar{q})$ is a $(\gamma_{\mathcal{P}} + \gamma_{\mathcal{Q}})$-approximate minimax equilibrium of the game.*

**Lemma 3.1.** *For an $\epsilon$-approximate equilibrium of the C-bounded minimax game $(f, \lambda)$. For the original unbounded constraint optimization problem Eq. (2), we have that $\mathbb{E}_{h \sim f} \mathbb{E}_{\mathcal{D}}[u(\boldsymbol{a}, y) \cdot b(h(x), \boldsymbol{a})] \geq \mathrm{OPT}(\mathcal{H}, \mathcal{D}, \gamma) - 2\epsilon$, and $\mathrm{DecCE}(f) \leq \gamma + \frac{1 + 2\epsilon}{C}$.*

*Proof.* We use $f^*$ to denote the optimal feasible solution to Eq. (2), since the constraints are satisfied, we have that $\mathcal{L}(f^*, \hat{\lambda}) \leq \mathrm{OPT}(\mathcal{H}, \mathcal{D}, \gamma)$. We prove the theorem by considering two cases.

First, if $\hat{f}$ is a feasible solution to the problem Eq. (2), i.e. $\mathrm{DecCE}(\hat{f}) \leq \gamma \leq \gamma + \frac{1 + 2\epsilon}{C}$. Since $(\hat{f}, \bar{\lambda})$ is a $\epsilon$-approximate equilibrium, we have that

$$-\mathbb{E}_{h \sim f} \mathbb{E}_D \Big[ \sum_{\boldsymbol{a} \in \mathcal{A}} u(\boldsymbol{a}, y) \cdot b(h(x), \boldsymbol{a}) \Big] = \max_{\lambda} \mathcal{L}(\hat{f}, \lambda)$$

$$\leq \mathcal{L}(\hat{f}, \bar{\lambda}) + \epsilon$$

$$\leq \min_{f \in \Delta(\mathcal{H})} \mathcal{L}(\hat{f}, \bar{\lambda}) + 2\epsilon$$

$$\leq \mathcal{L}(f^*, \bar{\lambda}) + 2\epsilon$$

$$\leq -\mathbb{E}_{h \sim f^*} \mathbb{E}_D \Big[ \sum_{\boldsymbol{a} \in \mathcal{A}} u(\boldsymbol{a}, y) \cdot b(h(x), \boldsymbol{a}) \Big] + 2\epsilon$$

$$= -\mathrm{OPT}(\mathcal{H}, \mathcal{D}, \gamma).$$

Therefore, we have

$$\mathbb{E}_{h \sim f} \mathbb{E}_D \Big[ \sum_{\boldsymbol{a} \in \mathcal{A}} u(\boldsymbol{a}, y) \cdot b(h(x), \boldsymbol{a}) \Big] \geq \mathrm{OPT}(\mathcal{H}, \mathcal{D}, \gamma) - 2\epsilon.$$

Second, we consider the case where $\hat{f}$ is not a feasible solution to Eq. (2). Let $(\hat{s}, \hat{i}, \hat{j}, \hat{a}_i) = \arg\max_{s,i,j,a_i} s(\mathbb{E}_f \mathbb{E}_D[(h(x)_j - y_j) \cdot b_i(h(x), a_i)] - \gamma) > 0$, and let $\lambda'$ be the vector such that the $(\hat{s}, \hat{i}, \hat{j}, \hat{a}_i)$-th coordinate $\lambda'_{\hat{s}, \hat{i}, \hat{j}, \hat{a}_i} = C$, and all else coordinates are 0, then we have that given $\lambda' \in \arg\max_\lambda \mathcal{L}(\hat{f}, \lambda)$. Therefore, since $(\hat{f}, \bar{\lambda})$ is a $\epsilon$-approximate equilibrium, we have

$$\mathcal{L}(\hat{f}, \bar{\lambda}) \geq \max_\lambda \mathcal{L}(\hat{f}, \lambda) - \epsilon$$

$$= -\mathbb{E}_{h \sim f} \mathbb{E}_D [\sum_{\boldsymbol{a} \in \mathcal{A}} u(\boldsymbol{a}, y) \cdot b(h(x), \boldsymbol{a})] + C\hat{s}\Big(\mathbb{E}_f \mathbb{E}_D[(h(x)_{\hat{j}} - y_{\hat{j}}) \cdot b_{\hat{i}}(h(x), a_{\hat{i}})] - \gamma\Big) - \epsilon$$

Therefore,

$$- \mathbb{E}_{h \sim f} \mathbb{E}_D [\sum_{\boldsymbol{a} \in \mathcal{A}} u(\boldsymbol{a}, y) \cdot b(h(x), \boldsymbol{a})] + C\hat{s}\Big(\mathbb{E}_f \mathbb{E}_D[(h(x)_{\hat{j}} - y_{\hat{j}}) \cdot b_{\hat{i}}(h(x), a_{\hat{i}})] - \gamma\Big)$$

$$\leq \mathcal{L}(\hat{f}, \bar{\lambda}) + \epsilon$$
$$\leq \mathcal{L}(f^*, \bar{\lambda}) + 2\epsilon$$
$$\leq -\mathbb{E}_{h \sim \hat{f}} \mathbb{E}_D [\sum_{\boldsymbol{a} \in \mathcal{A}} u(\boldsymbol{a}, y) \cdot b(h(x), \boldsymbol{a})] + 2\epsilon$$

Since $\forall \boldsymbol{a}, y, u(\boldsymbol{a}, y) \in [0, 1]$, we have that

$$C\hat{s}\Big(\mathbb{E}_f \mathbb{E}_D[(h(x)_{\hat{j}} - y_{\hat{j}}) \cdot b_{\hat{i}}(h(x), a_{\hat{i}})] - \gamma\Big) \leq 1 + 2\epsilon.$$

Thus,

$$\max_{s,i,j,a_i} s(\mathbb{E}_f \mathbb{E}_D[(h(x)_j - y_j) \cdot b_i(h(x), a_i)] - \gamma) \leq \frac{1 + 2\epsilon}{C},$$

and this implies that $\mathrm{DecCE}(\hat{f}) \leq \gamma + \frac{1+2\epsilon}{C}$. $\qquad\square$

We need the following technical lemmas before proving Lemma 3.2. For any function $\psi : \mathcal{X} \times \mathcal{Y} \to \mathbb{R}$ and any dataset $D = \{(x^{(i)}, y^{(i)})\}_{i=1}^n$, we denote the empirical expectation of $\psi$ over $D$ as

$$\hat{\mathbb{E}}_D[\psi(x, y)] \triangleq \frac{1}{n} \sum_{i=1}^n \psi(x^{(i)}, y^{(i)}).$$

**Theorem H.1.** *Fix a finite-size class of deterministic predictors $\mathcal{H}$. For any distribution $\mathcal{D}$, let $D \sim \mathcal{D}^n$ be a dataset consisting of $n$ samples $(x^{(i)}, y^{(i)})$ sampled i.i.d. from $\mathcal{D}$. Then for any $\delta \in (0, 1)$, with probability $1 - \delta$, for every $f \in \Delta(\mathcal{H})$, we have*

$$\left| \mathbb{E}_\mathcal{D} \mathbb{E}_{h \sim f} \left[ \sum_{\boldsymbol{a} \in \mathcal{A}} u(\boldsymbol{a}, y) \cdot b(h(x), \boldsymbol{a}) \right] - \hat{\mathbb{E}}_D \mathbb{E}_{h \sim f} \left[ \sum_{\boldsymbol{a} \in \mathcal{A}} u(\boldsymbol{a}, y) \cdot b(h(x), \boldsymbol{a}) \right] \right| \leq \sqrt{\frac{\ln \frac{2|\mathcal{H}|}{\delta}}{2n}}.$$

*Proof.* For any $(x, y) \sim \mathcal{D}$, observe that $\sum_{\boldsymbol{a} \in \mathcal{A}} u(\boldsymbol{a}, y) \cdot b(h(x), \boldsymbol{a}) \leq \sum b(h(x), \boldsymbol{a}) = 1$. By Hoeffding's inequality, we have for any $h \in \mathcal{H}$, for any $\delta' \in (0, 1)$ with probability $1 - \delta'$, we have

$$\left| \mathbb{E}_\mathcal{D} \left[ \sum_{\boldsymbol{a} \in \mathcal{A}} u(\boldsymbol{a}, y) \cdot b(h(x), \boldsymbol{a}) \right] - \hat{\mathbb{E}}_D \left[ \sum_{\boldsymbol{a} \in \mathcal{A}} u(\boldsymbol{a}, y) \cdot b(h(x), \boldsymbol{a}) \right] \right| \leq \sqrt{\frac{\ln \frac{2}{\delta'}}{2n}}.$$

Then let $\delta' = \delta/|\mathcal{H}|$, by union bound we have with probability $1 - \delta$, for any $h \in \mathcal{H}$,

$$\left| \mathbb{E}_\mathcal{D} \left[ \sum_{\boldsymbol{a} \in \mathcal{A}} u(\boldsymbol{a}, y) \cdot b(h(x), \boldsymbol{a}) \right] - \hat{\mathbb{E}}_D \left[ \sum_{\boldsymbol{a} \in \mathcal{A}} u(\boldsymbol{a}, y) \cdot b(h(x), \boldsymbol{a}) \right] \right| \leq \sqrt{\frac{\ln \frac{2|\mathcal{H}|}{\delta}}{2n}}.$$

Finally we have with probability $1 - \delta$, for any $f \in \Delta(\mathcal{H})$,

$$\left| \mathbb{E}_{\mathcal{D}} \mathbb{E}_{h \sim f} \left[ \sum_{\boldsymbol{a} \in \mathcal{A}} u(\boldsymbol{a}, y) \cdot b(h(x), \boldsymbol{a}) \right] - \hat{\mathbb{E}}_D \mathbb{E}_{h \sim f} \left[ \sum_{\boldsymbol{a} \in \mathcal{A}} u(\boldsymbol{a}, y) \cdot b(h(x), \boldsymbol{a}) \right] \right|$$

$$= \left| \mathbb{E}_{h \sim f} \left[ \mathbb{E}_{\mathcal{D}} \left[ \sum_{\boldsymbol{a} \in \mathcal{A}} u(\boldsymbol{a}, y) \cdot b(h(x), \boldsymbol{a}) \right] \right] - \mathbb{E}_{h \sim f} \left[ \hat{\mathbb{E}}_D \left[ \sum_{\boldsymbol{a} \in \mathcal{A}} u(\boldsymbol{a}, y) \cdot b(h(x), \boldsymbol{a}) \right] \right] \right|$$

$$= \left| \mathbb{E}_{h \sim f} \left[ \mathbb{E}_{\mathcal{D}} \left[ \sum_{\boldsymbol{a} \in \mathcal{A}} u(\boldsymbol{a}, y) \cdot b(h(x), \boldsymbol{a}) \right] - \hat{\mathbb{E}}_D \left[ \sum_{\boldsymbol{a} \in \mathcal{A}} u(\boldsymbol{a}, y) \cdot b(h(x), \boldsymbol{a}) \right] \right] \right|$$

$$\leq \mathbb{E}_{h \sim f} \left[ \left| \mathbb{E}_{\mathcal{D}} \left[ \sum_{\boldsymbol{a} \in \mathcal{A}} u(\boldsymbol{a}, y) \cdot b(h(x), \boldsymbol{a}) \right] - \hat{\mathbb{E}}_D \left[ \sum_{\boldsymbol{a} \in \mathcal{A}} u(\boldsymbol{a}, y) \cdot b(h(x), \boldsymbol{a}) \right] \right| \right]$$

$$\leq \mathbb{E}_{h \sim f} \left[ \sqrt{\frac{\ln \frac{2|\mathcal{H}|}{\delta}}{2n}} \right] = \sqrt{\frac{\ln \frac{2|\mathcal{H}|}{\delta}}{2n}}.$$

$\square$

**Theorem H.2.** *Fix a finite-size class of deterministic predictors $\mathcal{H}$. For any distribution $\mathcal{D}$, let $D \sim \mathcal{D}^n$ be a dataset consisting of $n$ samples $(x^{(i)}, y^{(i)})$ sampled i.i.d. from $\mathcal{D}$. Then for any $\delta \in (0, 1)$, with probability $1 - \delta$, for every $f \in \Delta(\mathcal{H}), i \in [N], j \in [d], a_i \in \mathcal{A}_i$ we have*

$$\left| \mathbb{E}_{\mathcal{D}} \mathbb{E}_{h \sim f}[(y_j - h(x)_j) \cdot b_i(h(x), a_i)] - \hat{\mathbb{E}}_D \mathbb{E}_{h \sim f}[(y_j - h(x)_j) \cdot b_i(h(x), a_i)] \right|$$

$$\leq \sqrt{\frac{8 \ln \frac{2|\mathcal{H}|Ndm}{\delta}}{n}}.$$

*Proof.* For any $(x, y) \sim \mathcal{D}, i \in [N], j \in [d]$, observe that $(y_j - h(x)_j) \cdot b_i(h(x), a_i) \in [-2, 2]$. By Hoeffding's inequality, we have for any $h \in \mathcal{H}$, for any $\delta' \in (0, 1)$ with probability $1 - \delta'$, we have

$$\left| \mathbb{E}_{\mathcal{D}}[(y_j - h(x)_j) \cdot b_i(h(x), a_i)] - \hat{\mathbb{E}}_D[(y_j - h(x)_j) \cdot b_i(h(x), a_i)] \right| \leq \sqrt{\frac{8 \ln \frac{2}{\delta'}}{n}}.$$

Then let $\delta' = \delta/(|\mathcal{H}|Ndm)$, by union bound we have with probability $1 - \delta$, for any $h \in \mathcal{H}, i \in [N]$ and $j \in [d]$ and $a_i \in \mathcal{A}_i$

$$\left| \mathbb{E}_{\mathcal{D}}[(y_j - h(x)_j) \cdot b_i(h(x), a_i)] - \hat{\mathbb{E}}_D[(y_j - h(x)_j) \cdot b_i(h(x), a_i)] \right| \leq \sqrt{\frac{8 \ln \frac{2|\mathcal{H}|Ndm}{\delta}}{n}}.$$

Finally we have with probability $1 - \delta$, for any $f \in \Delta(\mathcal{H})$,

$$\left| \mathbb{E}_{(x,y) \sim \mathcal{D}} \mathbb{E}_{h \sim f}[(y_j - h(x)_j) \cdot b_i(h(x), a_i)] - \hat{\mathbb{E}}_D \mathbb{E}_{h \sim f}[(y_j - h(x)_j) \cdot b_i(h(x), a_i)] \right|$$

$$= \left| \mathbb{E}_{h \sim f} \left[ \mathbb{E}_{(x,y) \sim \mathcal{D}}[(y_j - h(x)_j) \cdot b_i(h(x), a_i)] \right] - \mathbb{E}_{h \sim f} \left[ \hat{\mathbb{E}}_D[(y_j - h(x)_j) \cdot b_i(h(x), a_i)] \right] \right|$$

$$= \left| \mathbb{E}_{h \sim f} \left[ \mathbb{E}_{(x,y) \sim \mathcal{D}}[(y_j - h(x)_j) \cdot b_i(h(x), a_i)] - \hat{\mathbb{E}}_D[(y_j - h(x)_j) \cdot b_i(h(x), a_i)] \right] \right|$$

$$\leq \mathbb{E}_{h \sim f} \left[ \left| \mathbb{E}_{(x,y) \sim \mathcal{D}}[(y_j - h(x)_j) \cdot b_i(h(x), a_i)] - \hat{\mathbb{E}}_D[(y_j - h(x)_j) \cdot b_i(h(x), a_i)] \right| \right]$$

$$\leq \mathbb{E}_{h \sim f} \left[ \sqrt{\frac{8 \ln \frac{2|\mathcal{H}|Ndm}{\delta}}{n}} \right] = \sqrt{\frac{8 \ln \frac{2|\mathcal{H}|Ndm}{\delta}}{n}}.$$

$\square$

Now we are ready to prove Lemma 3.2.

**Lemma 3.2.** *We have* $|\mathcal{L}_{\mathcal{D}}(f, \lambda) - \mathcal{L}_D(f, \lambda)| \leq \sqrt{\frac{\ln \frac{|4\mathcal{H}|}{\delta}}{2n}} + C\sqrt{\frac{8\ln \frac{4|\mathcal{H}|Ndm}{\delta}}{n}}$ *for all* $f \in \Delta(\mathcal{H}), \lambda \in$ $\Lambda$ *with probability* $1 - \delta$.

*Proof.* This is straightforward from Theorem H.1 and Theorem H.2, we split the budget to $\delta/2$,

$$\mathcal{L}_D(f, \lambda) - \mathcal{L}_{\mathcal{D}}(f, \lambda)$$

$$= -\mathbb{E}_{h \sim f}\mathbb{E}_{\mathcal{D}}\Big[\sum_{\boldsymbol{a} \in \mathcal{A}} u(\boldsymbol{a}, y) \cdot b(h(x), \boldsymbol{a})\Big] + -\mathbb{E}_{h \sim f}\mathbb{E}_D\Big[\sum_{\boldsymbol{a} \in \mathcal{A}} u(\boldsymbol{a}, y) \cdot b(h(x), \boldsymbol{a})\Big]$$

$$+ \sum_{s \in \{+,-\}} \sum_{i=1}^{N}\sum_{j=1}^{d} \sum_{a_i \in A_i} \lambda_{s,i,j,a_i} s(\mathbb{E}_f\mathbb{E}_{\mathcal{D}}[(h(x)_j - y_j) \cdot b_i(h(x), a_i)] - \gamma)$$

$$- \sum_{s \in \{+,-\}} \sum_{i=1}^{N}\sum_{j=1}^{d} \sum_{a_i \in A_i} \lambda_{s,i,j,a_i} s(\mathbb{E}_f\mathbb{E}_{\mathcal{D}}[(h(x)_j - y_j) \cdot b_i(h(x), a_i)] - \gamma)$$

Therefore,

$$|\mathcal{L}_D(f, \lambda) - \mathcal{L}_{\mathcal{D}}(f, \lambda)|$$

$$\leq \Big|\mathbb{E}_{h \sim f}\mathbb{E}_{\mathcal{D}}\Big[\sum_{\boldsymbol{a} \in \mathcal{A}} u(\boldsymbol{a}, y) \cdot b(h(x), \boldsymbol{a})\Big] - \mathbb{E}_{h \sim f}\mathbb{E}_D\Big[\sum_{\boldsymbol{a} \in \mathcal{A}} u(\boldsymbol{a}, y) \cdot b(h(x), \boldsymbol{a})\Big]\Big| + \sum_{s \in \{+,-\}} \sum_{i=1}^{N}\sum_{j=1}^{d} \sum_{a_i \in A_i}$$

$$\lambda_{s,i,j,a_i} s|\mathbb{E}_f\mathbb{E}_{\mathcal{D}}[(h(x)_j - y_j) \cdot b_i(h(x), a_i)] - \mathbb{E}_f\mathbb{E}_D[(h(x)_j - y_j) \cdot b_i(h(x), a_i)]|$$

$$\leq \sqrt{\frac{\ln \frac{|4\mathcal{H}|}{\delta}}{2n}} + C\sqrt{\frac{8\ln \frac{4|\mathcal{H}|Ndm}{\delta}}{n}}.$$

$\square$

Finally we are ready to prove Theorem 3.1.

**Theorem 3.1.** *Suppose* `PerDecCal` *runs for* $T = O(\log(Nmd)/\epsilon^4)$ *rounds and is given a dataset* $D$ *drawn i.i.d. from* $\mathcal{D}$ *of size* $n \geq O(\frac{\log(|\mathcal{H}|Ndm/\delta)}{\epsilon^4})$. *With probability at least* $1 - \delta$, *the output predictor* $\hat{f}$ *that satisfies*

1. $\text{DecCE}(\hat{f}) \leq \gamma + \epsilon$.

2. *Suppose the receivers play strict best response to* $\hat{f}$. *Then the receivers obtain swap regret bounded by* $2mL(\gamma + \epsilon)$. *The sender achieves* $\epsilon$-*optimal utility:*
$$\mathbb{E}_{h \sim f}\mathbb{E}_{\mathcal{D}}[u(\boldsymbol{a}, y) \cdot b(h(x), \boldsymbol{a})] \geq \text{OPT}(\mathcal{H}, \mathcal{D}, \gamma) - \epsilon.$$

*Proof.* The regret bound of Hedge algorithm is $O(C\sqrt{T \log Nmd})$, and the best response of $f$ give non-positive regret. From Lemma H.1 and $T = O(\log(Nmd/\delta)/\epsilon^4)$, we have that $(\hat{f}, \bar{\lambda})$ is an $\varepsilon/4$-approximate equilibrium under $\mathcal{L}_D(f, \lambda)$. From Lemma 3.2, we know that with probability $1 - \delta$, $\forall f \in \Delta(\mathcal{H}), \lambda \in \Lambda$,

$$|\mathcal{L}_{\mathcal{D}}(f, \lambda) - \mathcal{L}_{\mathcal{D}}(f, \lambda)| \leq \sqrt{\frac{\ln \frac{|4\mathcal{H}|}{\delta}}{2n}} + C\sqrt{\frac{8\ln \frac{4|\mathcal{H}|Ndm}{\delta}}{n}}.$$

Therefore, let $f' = \arg\min_{f \in \Delta(\mathcal{H})} \mathcal{L}_{\mathcal{D}}(f, \bar{\lambda})$

$$\mathcal{L}_{\mathcal{D}}(\hat{f}, \bar{\lambda}) - \mathcal{L}_{\mathcal{D}}(f', \bar{\lambda}) \leq \mathcal{L}_{\mathcal{D}}(\hat{f}, \bar{\lambda}) - \mathcal{L}_D(\hat{f}, \bar{\lambda}) + \mathcal{L}_D(\hat{f}, \bar{\lambda}) - \mathcal{L}_D(f', \bar{\lambda}) + \mathcal{L}_D(f', \bar{\lambda}) - \mathcal{L}_{\mathcal{D}}(f', \bar{\lambda})$$

$$\leq \epsilon/2 + 2\sqrt{\frac{\ln \frac{|4\mathcal{H}|}{\delta}}{2n}} + 2C\sqrt{\frac{8\ln \frac{4|\mathcal{H}|Ndm}{\delta}}{n}}.$$

Similarly, we can prove that

$$\max_{\lambda \in \Lambda} \mathcal{L}_{\mathcal{D}}(\hat{f}, \lambda) - \mathcal{L}_{\mathcal{D}}(\hat{f}, \bar{\lambda}) \leq \epsilon/4 + 2\sqrt{\frac{\ln \frac{|4\mathcal{H}|}{\delta}}{2n}} + 2C\sqrt{\frac{8\ln \frac{4|\mathcal{H}|Ndm}{\delta}}{n}}.$$

Therefore, $(\hat{f}, \bar{\lambda})$ is an $\epsilon/4 + 2\sqrt{\frac{\ln \frac{|4\mathcal{H}|}{\delta}}{2n}} + 2C\sqrt{\frac{8\ln\frac{4|\mathcal{H}|Ndm}{\delta}}{n}}$-approximate equilibrium for the payoff $\mathcal{L}_{\mathcal{D}}(f, \lambda)$ under the true distribution $\mathcal{D}$. When $n \geq O(\frac{\log(|\mathcal{H}|Ndm)}{\epsilon^4})$, we have $(\hat{f}, \bar{\lambda})$ is a $\epsilon/2$-approximate equilibrium under the true distribution $\mathcal{D}$. Then, by Lemma 3.1, and the choice of $C$, we have

$$\text{DecCE}(\hat{f}) \leq \gamma + \frac{1+\epsilon}{C} \leq \gamma + \frac{1+\epsilon}{2/\epsilon} \leq \gamma + \frac{2}{2/\epsilon} = \gamma + \epsilon,$$

and

$$\mathbb{E}_{h\sim f}\mathbb{E}_{\mathcal{D}}[u(\boldsymbol{a}, y) \cdot b(h(x), \boldsymbol{a})] \geq \text{OPT}(\mathcal{H}, \mathcal{D}, \gamma) - \epsilon.$$

Finally by Theorem 2.1, the receivers who play $\eta$-quantal response obtain the stated swap regret bound. $\qquad\square$

## I  MISSING PROOFS IN SECTION 4

**Lemma 4.1.** *Consider a randomized predictor $f \in \Delta(\mathcal{H}_{\text{ALL}})$ where $\mathcal{H}_{\text{ALL}} = \{h : \mathcal{X} \to \mathcal{Y}\}$ is the class of all possible deterministic predictors. There exists a distribution $Q_f \in \Delta(\mathcal{Y})$ such that for any $v \in \mathcal{Y}$, $Q_f(v) = \text{Pr}_{h\sim f,(x,y)\sim\mathcal{D}}[h(x) = v]$. A distribution $Q \in \Delta(\mathcal{Y})$ corresponds to the distribution over posterior means induced by some signaling scheme if and only if it is the prediction distribution $Q_f$ of a perfectly calibrated predictor $f$.*

*Proof.* Fix any signaling scheme $\pi : \mathcal{Y} \to \Delta(S)$. Any context $x' \in \mathcal{X}$ corresponds to a state $\theta_{x'} = \mathbb{E}[y \mid x']$ and hence corresponds to a distribution of signals. Any signal $s \in S$ corresponds to a posterior mean $\mathbb{E}[\theta_{x'} \mid s]$. We define the mapping $g : \mathcal{X} \to \Delta(\mathcal{Y})$ such that

$$\text{Pr}[g(x) = \mathbb{E}[\theta_{x'} \mid s]] = \pi(s \mid \theta_x).$$

We have that for any $s \in S$

$$\mathbb{E}[y \mid g(x) = \mathbb{E}[\theta_{x'} \mid s]] = \int_y y \cdot \text{Pr}[y \mid g(x) = \mathbb{E}[\theta_{x'} \mid s]]dy$$

$$= \int_y y \cdot \left(\int_x \text{Pr}[y \mid x, s]dx\right)dy$$

$$= \int_x \left(\int_y \text{Pr}[y \mid x, s]dy\right)dx$$

$$= \int_x \mathbb{E}[y \mid x, s]dx$$

$$= \int_x \theta_x \text{Pr}[x \mid s]dx$$

$$= \mathbb{E}[\theta_{x'} \mid s]$$

Therefore, we can convert $g$ to a randomized predictor $f \in \Delta(\mathcal{H}_{\text{ALL}})$ that is perfectly calibrated.

Now consider any randomized predictor $f$ that is calibrated, i.e. we have that for any $v \in \mathcal{Y}$

$$\mathbb{E}_{h\sim f}\mathbb{E}_{(x,y)\sim\mathcal{D}}[y - h(x) \mid h(x) = v] = 0.$$

Now consider a signaling scheme such that the signal set is $\mathcal{Y}$ and given any state $\theta = \mathbb{E}[y \mid x]$, it sends signal $v \in \mathcal{Y}$ with probability $\text{Pr}_{h\sim f}[h(x) = v \mid \theta]$. We have that for any signal $v$, the posterior mean is

$$\mathbb{E}[\theta \mid v] = \mathbb{E}[\theta \mid h(x) = v]$$

$$= \mathbb{E}[\mathbb{E}[y \mid x] \mid h(x) = v]$$

$$= \mathbb{E}[y \mid h(x)]$$

$$= v.$$

$\qquad\square$

**Lemma 4.2.** *For any randomized predictor $f$ is perfectly decision calibrated, we can construct a randomized predictor $f'$ such that (i) $f'$ is perfectly calibrated; (ii) the sender obtains the same expected utility under $f$ and $f'$.*

*Proof.* Without loss of generality, we assume that for any $a$, event $\{b(h(x), a) = 1\}$ happens with non-zero probability, otherwise we could remove that action. Since $f$ is perfectly decision calibrated, we have

$$\max_{j \in [d]} \max_{a \in \mathcal{A}} \mathbb{E}_{h \sim f} \mathbb{E}_{(x,y) \sim \mathcal{D}}[(y_j - h(x)_j) \cdot b(h(x), a)] = 0.$$

Since $\Pr\{b(h(x), a) = 1\} > 0$, equivalently, we have

$$\mathbb{E}_{h \sim f} \mathbb{E}_{(x,y) \sim \mathcal{D}}[h(x) - y | b(h(x), a) = 1] = 0.$$

For any $a \in \mathcal{A}$, let $f'_a := \mathbb{E}_{h \sim f} \mathbb{E}_{(x,y) \sim \mathcal{D}}[h(x) | b(h(x), a) = 1]$ Consider a post-processing function $p(h(x)) = \sum_{a \in \mathcal{A}} b(h(x), a) f'_a$. Now we are ready to construct $f'$, we let $f' = p(f)$. Note that $f'$ only output at most $m$ values, we only need to check the level sets of each $f'_a$. Also, the set $\{y | b(y, a) = 1\}$ is convex, therefore $b(f'_a, a) = 1$.

$$\mathbb{E}_{h \sim f'} \mathbb{E}_{(x,y) \sim \mathcal{D}}[(y_j - h(x)_j) \cdot \mathbf{1}(h(x) = f'_a)]$$
$$= \mathbb{E}_{h \sim f'} \mathbb{E}_{(x,y) \sim \mathcal{D}}[(y_j - h(x)_j) \cdot b(h(x), a)]$$
$$= \mathbb{E}_{h \sim f} \mathbb{E}_{(x,y) \sim \mathcal{D}}[(y_j - p(h(x))_j) \cdot b(h(x), a)]$$
$$= \mathbb{E}_{h \sim f} \mathbb{E}_{(x,y) \sim \mathcal{D}}\left[\left(y_j - \left[\sum_{a' \in \mathcal{A}} b(h(x), a) f'_{a'}\right]_j\right) \cdot b(h(x), a)\right]$$
$$= \mathbb{E}_{h \sim f} \mathbb{E}_{(x,y) \sim \mathcal{D}}[(y_j - [b(h(x), a) f'_a]_j) \cdot b(h(x), a)]$$
$$= \mathbb{E}_{h \sim f} \mathbb{E}_{(x,y) \sim \mathcal{D}}[y_j \cdot b(h(x), a)] - \mathbb{E}_{h \sim f} \mathbb{E}_{(x,y) \sim \mathcal{D}}[(h(x)_j \cdot b(h(x), a)]$$
$$= 0.$$

$\square$

**Theorem 4.1.** *Given at least* $O(\frac{\ln(|\mathcal{H}| dm/\delta)}{\epsilon^4})$ *samples, with probability* $1 - \delta$, `PerDecCal` *can output a predictor* $\hat{f}$ *such that the expected sender's utility under* $\hat{f}$ *is no worse than* $\mathrm{BayesOPT}(\mu_{\mathcal{D}}, \Pi_{\mathcal{H}}) - \epsilon$ *where we denote the optimal sender utility under our Bayesian persuasion benchmark as* $\mathrm{BayesOPT}(\mu_{\mathcal{D}}, \Pi_{\mathcal{H}})$.

*Proof.* By Theorem 3.1 we know that given any tolerance $\gamma \geq 0$ and $n \geq O(\frac{\ln(|\mathcal{H}| dm/\delta)}{\epsilon^4})$, let $C = \frac{2}{\epsilon}$, when `PerDecCal` runs for $T = O(\ln(Nmd)/\epsilon^4)$ rounds, with probability at least $1 - \delta$, it outputs $\hat{f}$ that satisfies

$$\mathbb{E}_{h \sim f} \mathbb{E}_{\mathcal{D}}[u(\boldsymbol{a}, y) \cdot b(h(x), \boldsymbol{a})] \geq \mathrm{OPT}(\mathcal{H}, \mathcal{D}, \gamma) - \epsilon$$
$$\geq \mathrm{OPT}(\mathcal{H}, \mathcal{D}, 0) - \epsilon.$$

Note that $\mathrm{OPT}(\mathcal{H}, \mathcal{D}, 0)$ is the optimal sender utility achieved by the randomized predictors over $\mathcal{H}$ that are perfectly decision calibrated, i.e. $\mathcal{F}_{\mathrm{DCAL}}(\mathcal{H})$. By Definition 4.2 and Lemma 4.2, $\mathrm{OPT}(\mathcal{H}, \mathcal{D}, 0)$ is equal to the optimal sender utility achieved by the predictors in $\mathcal{F}_{\mathrm{CAL}}(\mathcal{H})$.

Then by Definition 4.2 and Lemma 4.1, $\mathrm{OPT}(\mathcal{H}, \mathcal{D}, 0)$ is equal to the optimal sender utility achieved by the signaling schemes in $\Pi_{\mathcal{H}}$, i.e. $\mathrm{BayesOPT}(\mu_{\mathcal{D}}, \Pi_{\mathcal{H}})$.

Putting all these together, we know that the sender's expected utility under $\hat{f}$ is greater than $\mathrm{BayesOPT}(\mu_{\mathcal{D}}, \Pi_{\mathcal{H}}) - \epsilon$.

$\square$

# J  ALGORITHM AND MISSING PROOFS IN SECTION 5

## J.1  THE ALGORITHM FOR INFINITE HYPOTHESIS CLASS

**Lagrangian and Minimax Game**  Similarly, we can introduce the Lagrangian and restrict the Lagrangian variables to be bounded as follows:

$$\min_{f \in \Delta\mathcal{H}} \max_{\lambda \in \Lambda} \tilde{\mathcal{L}}_{\mathcal{D}}(f, \lambda) := -\mathbb{E}_{h \sim f}\mathbb{E}_{\mathcal{D}}\Big[\sum_{\boldsymbol{a} \in \mathcal{A}} u(\boldsymbol{a}, y) \cdot \tilde{b}(h(x), \boldsymbol{a})\Big]$$

$$+ \sum_{s \in \{+, -\}} \sum_{i=1}^{N}\sum_{j=1}^{d} \sum_{a_i \in A_i} \lambda_{s,i,j,a_i} s\Big(\mathbb{E}_f\mathbb{E}_D[(h(x)_j - y_j) \cdot \tilde{b}_i(h(x), a_i)] - \gamma\Big). \tag{6}$$

Now we present our oracle efficient algorithm `SmPerDecCal`.

---

**Algorithm 2** `SmPerDecCal`

---

**Input:** A set of samples $D$, ERM oracle $\mathrm{ERM}(D, \lambda)$, dual bound $C$ and tolerance $\gamma$.
1: Initialize $\lambda_1 = \frac{C}{2Nmd}\mathbf{1}$.
2: **for** $t = 1, \cdots, T$ **do**
3:    Learner best responds to $\lambda_t$:
4:        Use the ERM oracle to compute $h_t = \mathrm{ERM}(D, \lambda_t)$.
5:    Auditor runs Hedge to obtain $\lambda_{t+1}$:
6:        $\lambda_{t+1} = \mathrm{Hedge}(c_{1:t})$ where $c_t(\lambda_{s,i,j,a_i}) = \lambda_{s,i,j,a_i} s(\mathbb{E}_D[(h_t(x) - y) \cdot \tilde{b}(h_t(x), a)] - \gamma)$.
7: **end for**
**Output:** Output $\hat{f} = \mathrm{Uniform}(h_1, \cdots, h_T)$.

---

## J.2  MISSING PROOFS

We need the following technical lemmas before proving Theorem 5.1.

**Lemma J.1** (Lipschitzness of single smoothed best response)**.** *For any* $i \in [N], a_i \in \mathcal{A}_i$, *we have that the function* $\tilde{b}_i(\cdot, a_i)$ *is* $2\eta L$*-Lipschitz in the* $L_\infty$ *norm.*

*Proof.* For any $z \in [-1, 1]^d$, for simplicity, we drop the subscript $i$. Let $g_a(z) \triangleq \exp(\eta v(z, a)), G(z) \triangleq \sum_{a'} \exp(\eta v(z, a'))$. We have

$$\nabla_z \tilde{b}(z, a) = \frac{\nabla_z g_a(z)}{G(z)} - \frac{g_a(z)\nabla_z G(z)}{G(z)^2}$$

$$= \frac{\eta g_a(z)}{G(z)}\nabla_z v(z, a) - \frac{\eta g_a(z)}{G(z)}\sum_{a'}\frac{g_{a'}(z)}{G(z)}\nabla_z v(z, a')$$

$$= \eta b(z, a)\nabla_z v(z, a) - \sum_{a'} b(z, a')\nabla_z v(z, a').$$

Therefore for any $z \in [-1, 1]^d$, we have

$$\left\|\nabla_z \tilde{b}(z, a)\right\|_1 \leq \eta\tilde{b}(z, a)\left(\|\nabla_z v(z, a)\|_1 + \sum_{a'}\tilde{b}(z, a')\|\nabla_z v(z, a')\|_1\right)$$

$$\leq \eta\tilde{b}(z, a)(L + L) \leq 2\eta L.$$

By mean-value theorem we have that

$$\left|\tilde{b}(z, a) - \tilde{b}(z', a)\right| = \left|\int_0^1 \nabla_z\tilde{b}(z' + t(z - z'), a) \cdot (z - z')dt\right|$$

$$\leq \sup_{t \in [0,1]} \left\| \nabla_z \tilde{b}(z' + t(z - z'), a) \right\|_1 \|z - z'\|_\infty$$

$$\leq 2\eta L \|z - z'\|_\infty.$$

$\square$

**Lemma J.2** (Lipschitzness of joint smoothed best response). *For any $\boldsymbol{a} \in \mathcal{A}$, the joint smoothed best response $\tilde{b}(\cdot, \boldsymbol{a})$ satisfies that*

$$\sum_{\boldsymbol{a} \in \mathcal{A}} \left| \tilde{b}(z, \boldsymbol{a}) - \tilde{b}(z', \boldsymbol{a}) \right| \leq 2\eta m N L \|z - z'\|_\infty.$$

*Proof.* For any predictions $z, z' \in [-1, 1]^d$ and joint action $\boldsymbol{a} = (a_1, \cdots, a_N)$, denote $b_i(z, a_i)$ as $u_i(a_i)$ and $b_i(z', a_i)$ as $v_i(a_i)$

$$\sum_{\boldsymbol{a} \in \mathcal{A}} |b(z, \boldsymbol{a}) - b(z', \boldsymbol{a})|$$

$$= \sum_{\boldsymbol{a} \in \mathcal{A}} \left| \prod_{i \in [N]} u_i(a_i) - \prod_{i \in [N]} v_i(a_i) \right|$$

$$= \sum_{\boldsymbol{a} \in \mathcal{A}} \left| \prod_{i \in [N]} u_i(a_i) - v_1(a_1) \prod_{i=2}^{N} u_i(a_i) + v_1(a_1) \prod_{i=2}^{N} u_i(a_i) - \prod_{i \in [N]} v_i(a_i) \right|$$

$$= \sum_{\boldsymbol{a} \in \mathcal{A}} \left| (u_1(a_1) - v_1(a_1)) \prod_{i=2}^{N} u_i(a_i) + v_1(a_1) \left( \prod_{i=2}^{N} u_i(a_i) - \prod_{i=2}^{N} v_i(a_i) \right) \right|$$

$$= \sum_{\boldsymbol{a} \in \mathcal{A}} \left| (u_1(a_1) - v_1(a_1)) \prod_{i=2}^{N} u_i(a_i) + v_1(a_1)(u_2(a_2) - v_2(a_2)) \prod_{i=3}^{N} u_i(a_i) \right.$$

$$\left. + v_1(a_1) v_2(a_2) \left( \prod_{i=3}^{N} u_i(a_i) - \prod_{i=3}^{N} v_i(a_i) \right) \right|$$

$$\vdots$$

$$= \sum_{\boldsymbol{a} \in \mathcal{A}} \left| \sum_{i=1}^{N} \prod_{j=1}^{i-1} v_j(a_j)(u_i(a_i) - v_i(a_i)) \prod_{j=i+1}^{N} u_j(a_j) \right|$$

$$\overset{(a)}{\leq} 2\eta L \|z - z'\|_\infty \sum_{\boldsymbol{a} \in \mathcal{A}} \left| \sum_{i=1}^{N} \prod_{j=1}^{i-1} v_j(a_j) \prod_{j=i+1}^{N} u_j(a_j) \right|$$

$$= 2\eta L \|z - z'\|_\infty \sum_{\boldsymbol{a} \in \mathcal{A}} \sum_{i=1}^{N} \prod_{j=1}^{i-1} v_j(a_j) \prod_{j=i+1}^{N} u_j(a_j)$$

$$= 2\eta L \|z - z'\|_\infty \sum_{i=1}^{N} \sum_{\boldsymbol{a} \in \mathcal{A}} \prod_{j=1}^{i-1} v_j(a_j) \prod_{j=i+1}^{N} u_j(a_j)$$

$$= 2\eta L \|z - z'\|_\infty \sum_{i=1}^{N} \sum_{a_i \in \mathcal{A}_i} \sum_{\boldsymbol{a}_{-i}} \prod_{j=1}^{i-1} v_j(a_j) \prod_{j=i+1}^{N} u_j(a_j)$$

$$\overset{(b)}{=} 2\eta L \|z - z'\|_\infty \sum_{i=1}^{N} \sum_{a_i \in \mathcal{A}_i} 1$$

$$= 2\eta m N L \|z - z'\|_\infty$$

where $(a)$ holds because of Lemma J.1 and $(b)$ holds because $\prod_{j=1}^{i-1} v_j(a_j) \prod_{j=i+1}^{N} u_j(a_j)$ can be viewed as the probability density function of $\boldsymbol{a}_{-i}$ and then $\sum_{\boldsymbol{a}_{-i}} \prod_{j=1}^{i-1} v_j(a_j) \prod_{j=i+1}^{N} u_j(a_j) = 1$. $\hfill\square$

**Lemma J.3.** *Fix a class of deterministic predictors $\mathcal{H}$. For any distribution $\mathcal{D}$, let $D \sim \mathcal{D}^n$ be a dataset consisting of $n$ samples $(x^{(i)}, y^{(i)})$ sampled i.i.d. from $\mathcal{D}$. Then for any $\delta \in (0,1)$, with $1 - \delta$, for every $f \in \Delta(\mathcal{H})$, we have*

$$\left| \mathbb{E}_{\mathcal{D}} \mathbb{E}_{h \sim f} \left[ \sum_{\boldsymbol{a} \in \mathcal{A}} u(\boldsymbol{a}, y) \cdot \tilde{b}(h(x), \boldsymbol{a}) \right] - \hat{\mathbb{E}}_D \mathbb{E}_{h \sim f} \left[ \sum_{\boldsymbol{a} \in \mathcal{A}} u(\boldsymbol{a}, y) \cdot \tilde{b}(h(x), \boldsymbol{a}) \right] \right|$$

$$\leq \inf_{\epsilon > 0} \left( 4\eta m N L \epsilon + \sqrt{\frac{\ln \frac{2\mathcal{N}(\mathcal{H}, d_\infty, \epsilon)}{\delta}}{2n}} \right).$$

*Proof.* Define $Z_h = |\mathbb{E}_{\mathcal{D}}[\sum_{\boldsymbol{a} \in \mathcal{A}} u(\boldsymbol{a}, y) \cdot \tilde{b}(h(x), \boldsymbol{a})] - \hat{\mathbb{E}}_D[\sum_{\boldsymbol{a} \in \mathcal{A}} u(\boldsymbol{a}, y) \cdot \tilde{b}(h(x), \boldsymbol{a})]|$. For any $h_1, h_2 \in \mathcal{H}$, we have

$$|Z_{h_1} - Z_{h_2}| = \left| \mathbb{E}_{\mathcal{D}} \left[ \sum_{\boldsymbol{a} \in \mathcal{A}} u(\boldsymbol{a}, y) \cdot \tilde{b}(h_1(x), \boldsymbol{a}) - \tilde{b}(h_2(x), \boldsymbol{a})) \right] \right.$$

$$\left. - \hat{\mathbb{E}}_D \left[ \sum_{\boldsymbol{a} \in \mathcal{A}} u(\boldsymbol{a}, y) \cdot (\tilde{b}(h_1(x), \boldsymbol{a}) - \tilde{b}(h_2(x), \boldsymbol{a})) \right] \right|$$

$$\overset{(a)}{\leq} \left| \mathbb{E}_{\mathcal{D}} \left[ \sum_{\boldsymbol{a} \in \mathcal{A}} u(\boldsymbol{a}, y) \cdot (\tilde{b}(h_1(x), \boldsymbol{a}) - \tilde{b}(h_2(x), \boldsymbol{a})) \right] \right|$$

$$+ \left| \hat{\mathbb{E}}_D \left[ \sum_{\boldsymbol{a} \in \mathcal{A}} u(\boldsymbol{a}, y) \cdot (\tilde{b}(h_1(x), \boldsymbol{a}) - \tilde{b}(h_2(x), \boldsymbol{a})) \right] \right|$$

$$\overset{(b)}{\leq} \mathbb{E}_{\mathcal{D}} \left[ \sum_{\boldsymbol{a}} \left| (\tilde{b}(h_1(x), \boldsymbol{a}) - \tilde{b}(h_2(x), \boldsymbol{a})) \right| \right]$$

$$+ \hat{\mathbb{E}}_D \left[ \sum_{\boldsymbol{a}} \left| (\tilde{b}(h_1(x), \boldsymbol{a}) - \tilde{b}(h_2(x), \boldsymbol{a})) \right| \right]$$

$$\overset{(c)}{\leq} 2\eta m N L (\mathbb{E}_{\mathcal{D}}[\|h_1(x) - h_2(x)\|_\infty] + \hat{\mathbb{E}}_D[\|h_1(x) - h_2(x)\|_\infty])$$

$$\overset{(d)}{\leq} 4\eta m N L \sup_{x \in \mathcal{X}} \|h_1(x) - h_2(x)\|_\infty,$$

where $(a)$ holds because of Triangle Inequality, $(b)$ holds because of $u(\boldsymbol{a}, y) \leq 1$, $(c)$ holds because of Lemma J.2 and $(d)$ holds by definition.

By Hoeffding's inequality, fixing any $h \in \mathcal{H}$, we have with probability $1 - \delta$,

$$|Z_h| \leq \sqrt{\frac{\ln \frac{2}{\delta}}{2n}}.$$

Then as a result of the standard covering number argument we conclude with probability $1 - \delta$ for any $h \in \mathcal{H}$,

$$|Z_h| \leq 4\eta m N L \epsilon + \sqrt{\frac{\ln \frac{2\mathcal{N}(\mathcal{H}, d_\infty, \epsilon)}{\delta}}{2n}}$$

where $d_\infty(h_1, h_2) \triangleq \sup_{x \in \mathcal{X}} \|h_1(x) - h_2(x)\|_\infty$. $\hfill\square$

**Lemma J.4.** *Fix a class of deterministic predictors $\mathcal{H}$. For any distribution $\mathcal{D}$, let $D \sim \mathcal{D}^n$ be a dataset consisting of $n$ samples $(x^{(i)}, y^{(i)})$ sampled i.i.d. from $\mathcal{D}$. Then for any $\delta \in (0,1)$, with probability $1 - \delta$, for every $f \in \Delta(\mathcal{H}), i \in [N], j \in [d], a_i \in \mathcal{A}_i$ we have*

$$\left| \mathbb{E}_{\mathcal{D}} \mathbb{E}_{h \sim f} \left[ (y_j - h(x)_j) \cdot \tilde{b}_i(h(x), a_i) \right] - \hat{\mathbb{E}}_D \mathbb{E}_{h \sim f} \left[ (y_j - h(x)_j) \cdot \tilde{b}_i(h(x), a_i) \right] \right|$$

$$\leq \inf_{\epsilon > 0} \left( 8 \eta L \epsilon + \sqrt{\frac{8 \ln \frac{2 \mathcal{N}(\mathcal{H}, d_\infty, \epsilon) N d m}{\delta}}{n}} \right).$$

*Proof.* Define $Z_h = \mathbb{E}_{\mathcal{D}}[(y_j - h(x)_j) \cdot \tilde{b}_i(h(x), a_i)] - \hat{\mathbb{E}}_D[(y_j - h(x)_j) \cdot \tilde{b}_i(h(x), a_i)]$. For any $h_1, h_2 \in \mathcal{H}$, we have

$$|Z_{h_1} - Z_{h_2}| = \left| \mathbb{E}_{\mathcal{D}} \left[ (y_j - h(x)_j) \cdot (\tilde{b}_i(h_1(x), a_i) - \tilde{b}_i(h_2(x), a_i)) \right] \right.$$

$$\left. - \hat{\mathbb{E}}_D \left[ (y_j - h(x)_j) \cdot (\tilde{b}_i(h_1(x), a_i) - \tilde{b}_i(h_2(x), a_i)) \right] \right|$$

$$\overset{(a)}{\leq} \left| \mathbb{E}_{\mathcal{D}} \left[ (y_j - h(x)_j) \cdot (\tilde{b}_i(h_1(x), a_i) - \tilde{b}_i(h_2(x), a_i)) \right] \right|$$

$$+ \left| \hat{\mathbb{E}}_D \left[ (y_j - h(x)_j) \cdot (\tilde{b}_i(h_1(x), a_i) - \tilde{b}_i(h_2(x), a_i)) \right] \right|$$

$$\overset{(b)}{\leq} 2 \left| \mathbb{E}_{\mathcal{D}} \left[ \tilde{b}_i(h_1(x), a_i) - \tilde{b}_i(h_2(x), a_i)) \right] \right| + 2 \left| \hat{\mathbb{E}}_D \left[ \tilde{b}_i(h_1(x), a_i) - \tilde{b}_i(h_2(x), a_i)) \right] \right|$$

$$\overset{(c)}{\leq} 4 \eta L \left( |\mathbb{E}_{\mathcal{D}}[\|h_1(x) - h_2(x)\|_\infty]| + \left| \hat{\mathbb{E}}_D[\|h_1(x) - h_2(x)\|_\infty] \right| \right)$$

$$\overset{(d)}{\leq} 8 \eta L \sup_{x \in \mathcal{X}} \|h_1(x) - h_2(x)\|_\infty$$

where $(a)$ holds because of Triangle Inequality, $(b)$ holds because of $y, h(x) \in [-1,1]^d$, $(c)$ holds because of Lemma J.1 and $(d)$ holds by definition.

By Hoeffding's inequality, fixing and $h \in \mathcal{H}$, we have with probability $1 - \delta$,

$$|Z_h| \leq \sqrt{\frac{8 \ln \frac{2}{\delta}}{n}}.$$

Then as a result of the standard covering number argument we conclude with probability $1 - \delta'$ for any $h \in \mathcal{H}$,

$$|Z_h| \leq 8 \eta L \epsilon + \sqrt{\frac{8 \ln \frac{2 \mathcal{N}(\mathcal{H}, d_\infty, \epsilon)}{\delta'}}{n}}$$

where $d_\infty(h_1, h_2) \triangleq \sup_{x \in \mathcal{X}} \|h_1(x) - h_2(x)\|_\infty$.

Then let $\delta' = \delta/(N d m)$, by union bound we have with probability $1 - \delta$, for any $h \in \mathcal{H}, i \in [N]$ and $j \in [d]$ and $a_i \in \mathcal{A}_i$,

$$\mathbb{E}_{\mathcal{D}} \left[ (y_j - h(x)_j) \cdot \tilde{b}_i(h(x), a_i) \right] - \hat{\mathbb{E}}_D \left[ (y_j - h(x)_j) \cdot \tilde{b}_i(h(x), a_i) \right] \leq 8 \eta L \epsilon + \sqrt{\frac{8 \ln \frac{2 \mathcal{N}(\mathcal{H}, d_\infty, \epsilon) N d m}{\delta}}{n}}.$$

$\square$

**Lemma J.5.** *With probability $1 - \delta$, $\forall f \in \Delta(\mathcal{H}), \lambda \in \Lambda$,*

$$\left| \tilde{\mathcal{L}}_D(f, \lambda) - \tilde{\mathcal{L}}_{\mathcal{D}}(f, \lambda) \right|$$

$$\leq \inf_{\epsilon > 0} \left( 4 \eta (mN + 2C) L \epsilon + \sqrt{\frac{\ln \frac{4 \mathcal{N}(\mathcal{H}, d_\infty, \epsilon)}{\delta}}{2n}} + C \sqrt{\frac{8 \ln \frac{4 \mathcal{N}(\mathcal{H}, d_\infty, \epsilon) N d m}{\delta}}{n}} \right).$$

*Proof.* This is straightforward from Lemma J.3 and Lemma J.4. First we have

$$\tilde{\mathcal{L}}_D(f,\lambda) - \tilde{\mathcal{L}}_{\mathcal{D}}(f,\lambda)$$

$$= -\mathbb{E}_{h \sim f}\mathbb{E}_{\mathcal{D}}\Big[\sum_{\boldsymbol{a} \in \mathcal{A}} u(\boldsymbol{a}, y) \cdot \tilde{b}(h(x), \boldsymbol{a})\Big] + -\mathbb{E}_{h \sim f}\mathbb{E}_D\Big[\sum_{\boldsymbol{a} \in \mathcal{A}} u(\boldsymbol{a}, y) \cdot \tilde{b}(h(x), \boldsymbol{a})\Big]$$

$$+ \sum_{s \in \{+,-\}} \sum_{i=1}^{N} \sum_{j=1}^{d} \sum_{a_i \in A_i} \lambda_{s,i,j,a_i} s\Big(\mathbb{E}_f\mathbb{E}_{\mathcal{D}}[(h(x)_j - y_j) \cdot \tilde{b}_i(h(x), a_i)] - \gamma\Big)$$

$$- \sum_{s \in \{+,-\}} \sum_{i=1}^{N} \sum_{j=1}^{d} \sum_{a_i \in A_i} \lambda_{s,i,j,a_i} s\Big(\mathbb{E}_f\mathbb{E}_{\mathcal{D}}[(h(x)_j - y_j) \cdot \tilde{b}_i(h(x), a_i)] - \gamma\Big)$$

Therefore, by union bound with probability $1 - \delta$,

$$\left|\tilde{\mathcal{L}}_D(f,\lambda) - \tilde{\mathcal{L}}_{\mathcal{D}}(f,\lambda)\right|$$

$$\leq \left|\mathbb{E}_{h \sim f}\mathbb{E}_{\mathcal{D}}\Big[\sum_{\boldsymbol{a} \in \mathcal{A}} u(\boldsymbol{a}, y) \cdot b(h(x), \boldsymbol{a})\Big] - \mathbb{E}_{h \sim f}\mathbb{E}_D\Big[\sum_{\boldsymbol{a} \in \mathcal{A}} u(\boldsymbol{a}, y) \cdot b(h(x), \boldsymbol{a})\Big]\right| + \sum_{s \in \{+,-\}} \sum_{i=1}^{N} \sum_{j=1}^{d} \sum_{a_i \in A_i}$$

$$\lambda_{s,i,j,a_i} s |\mathbb{E}_f\mathbb{E}_{\mathcal{D}}[(h(x)_j - y_j) \cdot b_i(h(x), a_i)] - \mathbb{E}_f\mathbb{E}_D[(h(x)_j - y_j) \cdot b_i(h(x), a_i)]|$$

$$\leq \inf_{\epsilon > 0} \left(4\eta(mN + 2C)L\epsilon + \sqrt{\frac{\ln\frac{4\mathcal{N}(\mathcal{H}, d_\infty, \epsilon)}{\delta}}{2n}} + C\sqrt{\frac{8\ln\frac{4\mathcal{N}(\mathcal{H}, d_\infty, \epsilon)Ndm}{\delta}}{n}}\right).$$

$\square$

Now we are ready to prove Theorem 5.1.

**Theorem 5.1.** *Suppose* `SmPerDecCal` *runs for $T = O(\log(Nmd)/\epsilon^4)$ rounds and is given a dataset $D$ drawn i.i.d of size $n \geq O(\ln\frac{\mathcal{N}(\mathcal{H}, d_\infty, \frac{\epsilon^2}{\eta L})Ndm}{\delta}/\epsilon^4)$. With probability at least $1 - \delta$, it outputs $\hat{f}$ that satisfies*

1. $\mathrm{SmDecCE}(\hat{f}) \leq \gamma + \epsilon.$

2. *Suppose the receivers play $\eta$-quantal response to $\hat{f}$. Then the receivers obtain swap regret bounded by $2mL(\gamma + \epsilon) + \frac{\ln m + 1}{\eta}$. The sender achieves $\epsilon$-optimal utility:*

$$\mathbb{E}_{h \sim f}\mathbb{E}_{\mathcal{D}}\Big[u(\boldsymbol{a}, y) \cdot \tilde{b}(h(x), \boldsymbol{a})\Big] \geq \mathrm{OPT}(\mathcal{H}, \mathcal{D}, \gamma) - \epsilon.$$

*Proof.* The regret bound of the Hedge algorithm is $O(C\sqrt{T \log Nmd})$, and the ERM oracle gives non-positive regret. From Lemma H.1 and $T = O(\log(Nmd)/\epsilon^4)$, we have that $(\hat{f}, \bar{\lambda})$ is an $\varepsilon/4$-approximate equilibrium under $\tilde{\mathcal{L}}_D(f,\lambda)$. From Lemma J.5, we know that when $n > \alpha(\epsilon, \delta, \eta, m, d, N, L) \triangleq O(\ln\frac{\mathcal{N}(\mathcal{H}, d_\infty, \frac{\epsilon^2}{\eta L})Ndm}{\delta}/\epsilon^4)$ with probability $1 - \delta$, $\forall f \in \Delta(\mathcal{H}), \lambda \in \Lambda$,

$$\left|\tilde{\mathcal{L}}_{\mathcal{D}}(f,\lambda) - \tilde{\mathcal{L}}_{\mathcal{D}}(f,\lambda)\right| \leq \frac{\epsilon}{8}.$$

Therefore, let $f' = \arg\min_{f \in \Delta(\mathcal{H})} \mathcal{L}_{\mathcal{D}}(f, \bar{\lambda})$

$$\tilde{\mathcal{L}}_{\mathcal{D}}(\hat{f}, \bar{\lambda}) - \tilde{\mathcal{L}}_{\mathcal{D}}(f', \bar{\lambda}) \leq \tilde{\mathcal{L}}_{\mathcal{D}}(\hat{f}, \bar{\lambda}) - \tilde{\mathcal{L}}_D(\hat{f}, \bar{\lambda})$$
$$+ \tilde{\mathcal{L}}_D(\hat{f}, \bar{\lambda}) - \tilde{\mathcal{L}}_D(f', \bar{\lambda}) + \tilde{\mathcal{L}}_D(f', \bar{\lambda}) - \tilde{\mathcal{L}}_{\mathcal{D}}(f', \bar{\lambda})$$
$$\leq \frac{\epsilon}{8} + \frac{\epsilon}{4} + \frac{\epsilon}{8} = \frac{\epsilon}{2}.$$

Similarly, we can prove that

$$\max_{\lambda \in \Lambda} \tilde{\mathcal{L}}_{\mathcal{D}}(\hat{f}, \lambda) - \tilde{\mathcal{L}}_{\mathcal{D}}(\hat{f}, \bar{\lambda}) \leq \frac{\epsilon}{2}.$$

Therefore, $(\hat{f}, \bar{\lambda})$ is an $\epsilon/2$-approximate equilibrium for the payoff $\mathcal{L}_{\mathcal{D}}(f, \lambda)$ under the true distribution $\mathcal{D}$. Then, similar to Lemma 3.1, we have

$$\mathrm{DecCE}(\hat{f}) \leq \gamma + \frac{1 + \epsilon}{C} \leq \gamma + \frac{1 + \epsilon}{2/\epsilon} \leq \gamma + \frac{2}{2/\epsilon} = \gamma + \epsilon,$$

and

$$\mathbb{E}_{h \sim f} \mathbb{E}_{\mathcal{D}}[u(\boldsymbol{a}, y) \cdot \tilde{b}(h(x), \boldsymbol{a})] \geq \mathrm{OPT}(\mathcal{H}, \mathcal{D}, \gamma) - \epsilon.$$

Finally by Theorem F.3, the receivers who play $\eta$-quantal response obtain the stated swap regret bound. $\qquad\square$

