# OpenReview forum: "Persuasive Prediction via Decision Calibration"
_ICLR.cc/2026/Conference — Submitted to ICLR 2026_

### Official Review · Reviewer_rkYE · 2025-10-25

**Soundness:** 3
**Presentation:** 2
**Contribution:** 3
**Rating:** 6
**Confidence:** 3

**Summary:**

This paper considers the sequential Information Design model, popularized by the Bayesian Persuasion model, but relaxes the assumption of a common prior. They motivate this from a prediction standpoint. The principal observes a feature vector x, with E[y|x] representing the "world state" a la persuasion. It is natural that without a rich amount of data, the sender/receiver would not have a joint prior over this. The principal chooses a classifier h(x), which serves as the "signal", and the follower best responds by essentially choosing the optimal action corresponding to the classification. The work focuses on decision-calibrated classifiers, under which the receiver optimal is to follow the recommendation. They then optimize for the optimal decision-calibrated mixture of classifiers (for the sender) by using a lagrangian/minimax approach. They relate this sender optima to that under the standard BP model.

I generally found the work to be quite interesting and well-motivated (once I understood the model). The common prior assumption in BP is generally quite restrictive and the authors make the right trade-offs to address this in a classification inspired setting. The use of decision calibrated classifiers, rigorous results on optimizing this, and their relation to the standard BP benchmark were well done.

The work could be improved by better explaining the model and clarifying a few things. The following questions came to mind for me:
- I understand the theoretical results consider an online setting where the principal chooses a decision, receiver best responds and the principal updates. It would help to make this online model explicit, especially when introducing theorem 3.1
- Can you spell out clearly the benefit of randomizing over the classifiers in this setting? In standard BP, randomizing induces different posteriors and the outcome can be interpreted from a concave closure perspective. Can you give some intuitions on randomization here?
- I see often the phrase "following the action recommendation". This is a bit imprecise here right? The signals sent by the principal is a real value h(x). This gets converted to an action using the br function. If it suffices to just recommend the action (and hide the h(x)) then why not explicitly mention the revelation principal. In general, the lack of revelation principle here is a bit confusing to me.

Lastly, and a deeper point, I found the fact that the principal choosing a single classifier to face n receivers quite interesting. The classical analogue of this in BP is public persuasion (On the Tractability of Public Persuasion with No Externalities, Haifeng Xu), which is known to be NP-Hard. Could you comment on the connection here, especially as it relates to section 4.

Happy to engage on this!

**Strengths:**

See above

**Weaknesses:**

See above

**Questions:**

See above

---

> ### Author Response · Authors · 2025-11-23
>
> Thank you for taking the time to review our submission! Please find our replies to your questions/comments below.
>
> **Q1 regarding the clarification on the setting:**
>
> Thank you for the question! The setting we consider in this paper is a distributional batch setting in the standard terminology: the principal does not update after the receiver best responds. Our main contribution is to remove the common prior assumption from Bayesian persuasion while remaining within the distributional batch setting.
>
> That said, it is true that our algorithms for learning such predictors and the corresponding analysis draw on techniques from online learning. We will clarify this connection more explicitly in the final version of the paper.
>
> **Q2 regarding the randomization over the classifier:**
>
> In the standard Bayesian persuasion literature, it is known that randomized signaling scheme gives higher sender utility comparing to the case only allowing deterministic signals. The analogous result holds here in this setting without common prior, randomized predictor can potentially gives higher sender utility. Also, allowing randomization over the hypothesis class makes the decision calibration error bilinear over the strategy space of both the min and max player, which is important for the algorithms and the minimax analysis.
>
> **Q3 regarding the action recommendation:**
>
> Thank you for pointing this out! In standard Bayesian persuasion, the revelation principle applies because the sender and receiver share a common prior, which allows the sender to reduce any signaling scheme to a direct mechanism that simply recommends an action. However, this reduction does not apply in our setting, where the prior cannot be accurately learned from finite past data.
>
> In this **prior-free machine learning** setting, the signal produced by the predictor is real-valued, and a more realistic behavioral model is that the receiver best responds to the prediction  $f(x)$, given that the predictor satisfies decision calibration. We will revise the phrasing to state this directly rather than using the shorthand “following the action recommendation’’ to avoid confusion.
>
>
> **Q4 regarding the $n$ receiver setting:**
>
> Thank you for raising this interesting question! The main focus of this paper is to remove the strong common-prior assumption in Bayesian persuasion. Unlike the classical Bayesian persuasion setting where the prior is known, we believe the more basic question in our setting is understanding the statistical complexity, which is what our proposed algorithms address.
>
> Regarding computational complexity, our results are consistent with Xu (2020): our algorithm is oracle-efficient, in the sense that it requires access to an ERM oracle. For certain hypothesis classes, such ERM problems are believed to be NP-hard in the worst case, which mirrors the computational barriers known in the common-prior Bayesian persuasion literature.
>
> We agree that understanding the computational complexity of this prior-free setting for both the single-receiver and multi-receiver settings is an interesting direction. We regard this as an open problem for future work.

---

### Official Review · Reviewer_gCE4 · 2025-10-27

**Soundness:** 3
**Presentation:** 3
**Contribution:** 3
**Rating:** 6
**Confidence:** 2

**Summary:**

The paper introduces persuasive prediction, a learning-based analogue of Bayesian persuasion that eliminates the need for a common-prior assumption. In this framework, a sender observes covariates, uses them to predict an outcome, and releases the prediction to influence receivers’ actions. The paper proposes decision calibration as the key behavioral proxy, which requires predictions to be unbiased conditional on the receiver’s best response and ensures that receivers incur no swap regret when reacting myopically. The authors present three main theoretical results. First, they prove the equivalence between decision calibration and the absence of swap regret. Second, they show the existence of an optimal decision-calibrated predictor whose utility equals that of a Bayesian sender with full prior knowledge in the single-receiver case. Third, they develop a statistically and computationally efficient algorithm that learns an
$\epsilon$-optimal decision-calibrated predictor with polynomial sample complexity independent of the covariate space.

**Strengths:**

Please find the strengths below:
1. The paper identifies an important limitation of Bayesian persuasion when the common prior is high-dimensional or difficult to observe. The proposed concept of decision calibration links calibration in machine learning with rational decision-making in game theory.
2. The authors prove that, in the single-receiver case, the optimal decision-calibrated predictor achieves the same utility as a Bayesian sender with full knowledge of the prior, demonstrating an equivalence to the Bayesian-optimal benchmark.
3. The proposed no-regret with ERM-oracle algorithm is computationally efficient, with sample complexity independent of the covariate dimension. Moreover, the framework flexibly incorporates stochastic receiver responses and randomized predictor classes.

**Weaknesses:**

Please find the weaknesses below:
1. The model assumes the existence of a true joint distribution $D(X,Y)$ that can be accurately learned. In practice, the predictor $f$ may be biased or only approximately estimated, making exact decision calibration difficult to satisfy.
2. The calibration constraint is an infinite family of conditional expectation constraints, which cannot be perfectly verified with finite samples. The paper does not analyze robustness under distribution shift or small-sample regimes.
3. The theoretical equivalence to Bayesian persuasion relies on perfect calibration and simplified outcome structures. Its validity under nonlinear dependencies between $X$ and $Y$ or in high-dimensional deep learning settings remains untested, and no empirical results are provided.

**Questions:**

The questions are related to the weaknesses:
1. Can similar optimality or equivalence results be derived for approximately calibrated predictors, where $f$ only satisfies calibration up to a bounded error?
2. Does a global equilibrium exist under decision calibration, and can convergence or stability be established beyond the no-swap-regret condition?
3. How robust is the proposed algorithm to distribution shift or model misspecification, and can finite-sample error bounds be extended to practical high-dimensional predictors such as deep networks?

---

> ### Author Response · Authors · 2025-11-23
>
> Thank you for taking the time to review our submission! Please find our replies to your questions/comments below.
>
> **W1 regarding estimating the joint distribution:**
>
> Thank you for the question! As noted in lines 207–209 of the paper, decision calibration is not difficult to satisfy. In particular, the **constant predictor** that outputs the mean $\mathbb{E}[Y]$ is itself decision-calibrated. This implies that any function class containing a cover of constant predictors will automatically ensure feasibility. Importantly, estimating the mean does **not** require learning the full joint distribution $\mathcal{D}$. It is a standard statistical quantity that can be accurately estimated from data using well-known concentration results.
>
> Moreover, rather than merely learning *any* decision-calibrated predictor, we focus on the more challenging task of **selecting an optimal in-class decision-calibrated predictor** with respect to the sender’s utility. For this stronger objective, Theorem 3.1 and Theorem 5.1 show that our approach remains **sample efficient**, again without needing to estimate the joint distribution $\mathcal{D}$.
>
>
>
> **W2 regarding the number of constraints and the sample complexity:**
>
> For clarification, the number of constraints in our definition of decision calibration is actually $2Nmd$, where $d$ is the dimension of the outcome space, $m$ is the number of actions, and $N$ is the number of receivers. Thus, the total number of constraints is **finite**. Moreover, we show in both Theorem 3.1 and Theorem 5.1 that our algorithms are **sample-efficient**. In addition, our experiments on a real-world dataset (which is relatively small) further demonstrate that the algorithm is empirically sample-efficient. See our response to W4.
>
> **W2&Q3 regarding the robustness under distribution shift:**
>
> The primary goal of this paper is to **remove the common-prior assumption** in the standard Bayesian persuasion framework, a setting in which there is also **no distribution shift**. We believe that even this more basic version of the problem—Bayesian persuasion *without* a common prior and distribution shift—has not been addressed in prior work.
>
> The question of how to provide trustworthy predictions under distribution shift is indeed an important and interesting direction. A promising approach may be to generalize decision calibration to a multi-group formulation. Following the observation made by Kim et al. (2022) [1], such a multi-group notion of calibration can ensure decision calibration under distribution shift when the density ratio between the source and target distributions is within the real-valued groups. We view this as an interesting question for future research.
>
>
> **W3&Q3 regarding the linearity assumptions and generalization to deep learning settings:**
>
> We would like to clarify that the linearity assumption we make is that $v_i(a, y) $ is **Lipschitz in $y$** for every action $a$. This assumption is
> **not** about the dependence structure between $x$ and $y$ in the data distribution, nor is it an assumption on the hypothesis class $\mathcal{H}$. Thus, our results directly apply to **much more complex hypothesis classes**, as long as the class has a **finite covering number** (a condition that holds for deep neural networks).
>
> Moreover, our additional experimental results indeed use **neural networks** as the underlying hypothesis class. Please see our response to W4 for further details.
>
>
> **W4 regarding the empirical evaluation:**
>
> We have included real-world experimental results on the FICO HELOC dataset. In summary, our algorithm achieves roughly 5× higher sender utility compared to standard ERM with cross-entropy loss, while maintaining a similar level of decision-calibration error.
> | Method                               | Sender utility ↑ | DC error ↓ |
> |--------------------------------------|------------------|------------|
> | ERM ( cross-entropy)            |        0.026    |  0.007     |
> | $\texttt{SmPerDecCal}$ |   0.163         |   0.008  |
>
> Further details can be found in our official comment titled “Experiment on Real-World Dataset”.

---

> ### Author Response · Authors · 2025-11-23
>
> **Q1 regarding the feasibility:**
>
> Thank you for the question! As mentioned in the paper, our results do indeed cover the case where the model class contains only an approximately feasible predictor. This is explicitly captured in Assumption 2.2.
>
> **Q2 regarding the global equilibrium:**
>
>  We would first like to clarify that the multi-receiver setting we study does not introduce any game-theoretic interaction among receivers, since each receiver’s utility is independent of the actions taken by others. However, the learned randomized predictor (together with the corresponding $\lambda$) does form an equilibrium of the underlying minimax game between the predictor (min player) and the auditor (max player).
>
> Regarding generalizations beyond swap regret: as noted in line 201 and in Appendix F, decision calibration also provides type-regret guarantees.
>
> **References:**
>
> [1] Kim M P, Kern C, Goldwasser S, et al. Universal adaptability: Target-independent inference that competes with propensity scoring[J]. Proceedings of the National Academy of Sciences, 2022, 119(4): e2108097119.

---

### Official Review · Reviewer_r3eE · 2025-10-31

**Soundness:** 3
**Presentation:** 3
**Contribution:** 3
**Rating:** 6
**Confidence:** 3

**Summary:**

The authors study the framework of persuasive calibration (introduced in [Feng & Tang 2025], which aims to model situations where a sender predicts a label that a group of receivers will then use to choose actions by best-responding. The authors focus on the setting where the sender seeks to incentivize certain actions but faces the challenge of not having a prior over the label y.
The authors study what happens when the learner selects from the set of calibrated predictors. In particular, they propose a sample-efficient algorithm to learn a randomized predictor f that maximizes the sender’s utility over the space of calibrated predictors.
They further show that, in the case of a single receiver, the utility achieved by their algorithm matches that of a fully informed Bayesian sender. Finally, they extend their framework by providing an approximately optimal algorithm for settings where receivers choose actions according to a softmax distribution rather than by strict best responses.

**Strengths:**

- The authors put effort to motivate the problem and give intuitive explanations for their assumptions
- The paper includes strong theoretical foundations. I appreciated the effort to gradually introduce the algorithm by guiding the reader to the working principles behind it using background on game theory and optimization.
- They make the connection of their algorithm perfomance on the 1-receiver case with the Bayesian case
- They introduce the smoothed version of best-responding so that they can study the challenging setting of infinite hypothesis class.

**Weaknesses:**

- In step 5 of both algorithm 1 and 2 the word "Auditor" is used but this is never explained. Is the auditor different from the learner and why?
- Absence of any empirical evaluation.

**Questions:**

- Can the writers think of a real-world dataset and define an experimental setup to evaluate their technical results. How does one choose tolerance gamma in practice?

---

> ### Author Response · Authors · 2025-11-23
>
> Thank you for taking the time to review our submission! Please find our replies to your questions/comments below.
>
> **W1 regarding the terminology**:
>
> Thank you for pointing this out! In our original terminology, the learner corresponds to the min player, as the min player updates the predictor at each round. Similarly, the auditor corresponds to the max player, as the max player assigns higher weight to the constraints on which the predictor incurs larger errors. The term auditor is drawn from the fairness and learning theory literature.
>
> For consistency and clarity, we will follow your suggestion and replace these terms with “min player” and “max player” in the pseudocode.
>
> **W2 and Q1 regarding the empirical evaluation**:
>
> We have included real-world experimental results on the FICO HELOC dataset. In summary, our algorithm achieves roughly 5× higher sender utility compared to standard ERM with cross-entropy loss, while maintaining a similar level of decision-calibration error.
> | Method                               | Sender utility ↑ | DC error ↓ |
> |--------------------------------------|------------------|------------|
> | ERM ( cross-entropy)            |        0.026    |  0.007     |
> | $\texttt{SmPerDecCal}$ |   0.163         |   0.008  |
>
> Further details can be found in our official comment titled “Experiment on Real-World Dataset”.

---

> ### Comment · Reviewer_r3eE · 2025-11-25
> **Experiment design does not capture theoretical setting**
>
> Thank you for taking time to review and update your work!
> Looking at your experiment description, I am not sure how representative it is of your setting. In particular, to my understanding you test only the setting with one receiver (the bank). In addition, it is not clear whether or why the provided predictions are calibrated or if/why the receiver adopt a quantal response.

---

> ### Author Response · Authors · 2025-11-26
>
> Thank you for the question. Yes, the reported decision-calibration error corresponds to the smoothed decision-calibration error in Definition 5.2. An error below $0.01$ is typically regarded as “approximately calibrated’’ in the calibration literature. This error can be reduced further by adjusting the dual bound $C$.
>
> Regarding the experimental setting: we used a single-receiver version in the main text for clarity, but this is also the natural choice for our motivating example. We will include more implementation details (e.g., the choice of hyperparameters) in the revision, but we previously decided to keep the rebuttal concise. In response to your question, we additionally evaluated a **three-receiver** version of our algorithm.
>
> In this extension, the sender’s utility is **additive** across the receivers: we use the same per-receiver utility as in the single-receiver case (large positive for approving a non-defaulting borrower, negative for approving a defaulting borrower, and zero for denials), and take a fixed weighted sum with weights $0.2, 0.3, 0.5$.
>
>
> Each receiver has its own linear utility over outcomes. Denials always yield zero utility, while approvals take the form  $v_i(y) = a_i y + b_i,$
> where different $(a_i,b_i)$ encode different degrees of conservatism.
> The three approval-utility functions we use are:
>
> | Receiver $i$ | Approval utility $v_i(y)$          |
> |--------------|------------------------------------|
> | 1            | $v_1(y) = -3.0y + 0.30$          |
> | 2            | $v_2(y) = -2.0y + 0.10$          |
> | 3            | $v_3(y) = -1.5y + 0.12$          |
>
> We again use an MLP predictor and include the same log-loss constraint in $\texttt{SmPerDecCal}$.
> On the HELOC test set, $\texttt{SmPerDecCal}$ improves sender utility by a lot while slightly reducing the smoothed decision-calibration error compared to ERM:
>
> | Method                       | Sender utility $\uparrow$ | DC error $\downarrow$ |
> |------------------------------|----------------------------|------------------------|
> | ERM (cross-entropy)      | 0.060                     | 0.007               |
> | $\texttt{SmPerDecCal}$ | 0.153                     | 0.005               |

---

### Official Review · Reviewer_Argb · 2025-11-04

**Soundness:** 3
**Presentation:** 2
**Contribution:** 3
**Rating:** 6
**Confidence:** 2

**Summary:**

The paper studies prior-free persuasion in a batch (iid) setting: a sender observes covariates $X$ and predicts $h(X)$ where $h\in \mathcal{H}$ is a class of predictors.   Receivers best respond, and the sender wants to maximize the utility subject to decision calibration: the predictions are calibrated conditional on the receiver's best response $b_i$ under some Lipschitz utility function.  The core algorithm casts the problem as a zero-sum game through Lagrangian if $\mathcal{H}$ is finite.  For infinite $\mathcal{H}$, they adopt the quantal response and the smoothed calibration notion.

**Strengths:**

- This is an active line of research relaxing the common prior in the mechanism design problem.
- The connection between calibrated prediction and signaling schemes is a nice way to understand persuasion.

**Weaknesses:**

- The algorithm results seem standard.  The reader expects more discussion on the optimality of the algorithm.
- Given there is a lot of work in the space, the reader feels the related work should be improved.  For instance, how to position the algorithmic contribution in multi-objective learning, e.g., Garg, Sumegha, et al.   In particular, can we view the best response function $b_i$ as some checking function, and decision calibration becomes multi-calibration to those functions?  Additionally, how the sample complexity compares to other work is not discussed.
- Instead of quantal responses, can the result be generalized to other smooth response functions, e.g., Lipschitz approximate best-response by Foster and Hart?

Garg, Sumegha, et al. "Oracle efficient online multicalibration and omniprediction." _Proceedings of the 2024 Annual ACM-SIAM Symposium on Discrete Algorithms (SODA)_. Society for Industrial and Applied Mathematics, 2024.

Foster, Dean P., and Sergiu Hart. "Smooth calibration, leaky forecasts, finite recall, and nash dynamics." _Games and Economic Behavior_ 109 (2018): 271-293.

**Questions:**

Can we view the problem as multi-calibration problem in Garg et al.?

Is the algorithm has better (computational/sample) complexity than existing works?

Can the result be generalized to other smooth response functions

---

> ### Author Response · Authors · 2025-11-23
>
> Thank you for taking the time to review our submission! Please find our replies to your questions/comments below.
>
> **W1&W2 regarding the optimality of the sample complexity and the related work:**
>
> Thank you for bringing up this question!  To the best of our knowledge, our algorithm is not directly comparable to existing methods. Prior work focuses on learning (decision)-calibrated predictors, whereas our goal is to learn an optimal in-class decision-calibrated predictor. Because of this distinction, the sample complexity guarantees established in our paper are not directly comparable to those in earlier literature.
>
> That said, understanding the optimality of our algorithm and establishing corresponding lower bounds for this problem remains an interesting direction for future research. We will leave this as an open question.
>
>
> **W3 regarding generalization of the quantal response:**
>
> Thank you for bringing up this question! The result can indeed be generalized to any $L$-Lipschitz response function. In our analysis, the only property of the quantal response that we rely on is its **Lipschitz continuity**.
>
> If such a Lipschitz decision rule also guarantees **near-optimal utility** when responding to the true outcomes compared to the utility induced by the strict best-response rule, then we can establish no-swap-regret guarantees for the receivers analogous to those in bullet point 2 of Theorem 5.1.
>
> We will incorporate this generalization into the final version of the paper.
>
>
>
> **Q1 regarding the multi-calibration algorithm:**
>
> The problem we study is **not** a special case of the multi-calibration setting in Garg et al. (2024). First, as we noted in our response to W1 & W2, our focus is on **selection among a class of decision-calibrated predictors**, whereas Garg et al. (2024) do **not** address the problem of learning or selecting an *optimal* multi-calibrated predictor.
>
> Second, our setting does **not** involve multi-group guarantees. The receivers’ decisions depend solely on the prediction $f(x)$, not on the context $x$ itself. Moreover, decision calibration is a weaker notion than full calibration, yet it is already sufficient to guarantee that receivers best-responding to the prediction incur no swap regret. Importantly, this does not mean we are pursuing an “easier” objective. Selecting an optimal predictor from a *larger* hypothesis class is not necessarily easier than selecting from a smaller one, especially since prior work does not address the **selection problem** at all.
>
>
> **Q2 regarding the sample complexity comparision:**
>
>  As noted in our response to W2, the focus of this paper fundamentally differs from existing work: we study optimization within the class of decision-calibrated predictors, whereas prior work typically focuses on learning a (decision-)calibrated predictor rather than selecting an optimal one. Because of this difference in objectives, the sample-complexity results are not directly comparable.
>
> **Q3 regarding generalization of the quantal response:**
>
>  See our response to W3.

---

### Author Response · Authors · 2025-11-23
**Experiment on Real-World Dataset**

To better connect our setting to a realistic lending application, we run an experiment on the public **FICO HELOC dataset**, which contains de-identified home-equity line of credit applications with a binary “RiskPerformance” label indicating whether the borrower became seriously delinquent. We model a small bank that receives a probability-of-default forecast from a predictor and then decides whether to approve or deny the loan. In this setting it is natural that the bank (receiver) is conservative: it approves only when the predicted default probability is sufficiently low, while the predictor (sender) prefers loans to be approved as long as they are unlikely to default.

We encode these preferences with simple linear utilities over actions and outcomes. The sender receives a large positive payoff for approving a non-defaulting borrower and a negative payoff for approving a defaulting borrower, while denials have utility zero:

| Sender utility      | y = 0 (no default)          | y = 1 (default)              |
|---------------------|-----------------------------|------------------------------|
| approve             | 2.5       | -0.5              |
| deny                | 0                         | 0                          |

The receiver’s utility is also linear: it gives a small positive payoff for approving a non-defaulting borrower and a large negative payoff for approving a defaulting borrower, while denials have utility 0:

| Receiver utility    | y = 0 (no default)          | y = 1 (default)              |
|---------------------|-----------------------------|------------------------------|
| approve             | 0.1   | -1.9        |
| deny                | 0      | 0                  |

On this dataset we compare standard cross-entropy ERM with $\texttt{SmPerDecCal}$ using an MLP function class. In addition, we introduce a log-loss constraint enforced via an extra dual variable in the Lagrangian; in practice this helps the sender’s strategy converge faster while achieving higher sender utility and lower decision-calibration error. Empirically, $\texttt{SmPerDecCal}$ (with the log-loss constraint) achieves 5× higher sender utility than ERM while maintaining comparable decision-calibration error. The test-set results (over 3 random seeds) are summarized below:

| Method                               | Sender utility ↑ | DC error ↓ |
|--------------------------------------|------------------|------------|
| ERM ( cross-entropy)            |        0.026    |  0.007     |
| $\texttt{SmPerDecCal}$ |   0.163         |   0.008  |

We will include this result into the final version of the paper.

---

### Meta-Review · Area_Chair_jiEN · 2025-12-23

**Summary:**

The paper proposes a learning-based framework for Bayesian persuasion without a common prior, termed "Persuasive Prediction".  The authors provide oracle-efficient algorithms (PerDecCal/SmPerDecCal) based on minimax duality and no-regret learning

Strengths:
- Nice link between calibrated prediction (ML) and signaling schemes (Econ).
- Analysis of sample complexity and oracle efficiency.

Weaknesses:
- All reviewers noted the complete lack of empirical evaluation in the initial submission. The rebuttal partially addressed this, but the provided experiments are very basic
- The paper would also benefit from re-writing to highlight the novelty of the approach.

Overall, the paper is an interesting contribution but in the current form is slightly below the ICLR acceptance bar.

**Reviewer Concerns:**

I think that the novelty and the model concerns raised by the reviewers are only partially addressed in the rebuttal

**Reviewer Scores:**

I think that the reviewers were not very likely to raise their scores.

---

### Decision · Program_Chairs · 2026-01-26

Reject